# *S. pombe wtf* drivers use dual transcriptional regulation and selective protein exclusion from spores to cause meiotic drive

Nicole L. Nuckolls[1☯¤a], Ananya Nidamangala Srinivasa[1,2☯], Anthony C. Mok[1,3], Rachel M. Helston[1], María Angélica Bravo Núñez[1¤b], Jeffrey J. Lange[1], Todd J. Gallagher[1], Chris W. Seidel[1], Sarah E. Zanders[1,2]*

**1** Stowers Institute for Medical Research, Kansas City, Missouri, United States of America, **2** Department of Cell Biology and Physiology, University of Kansas Medical Center, Kansas City, Kansas, United States of America, **3** University of Missouri—Kansas City, Kansas City, Missouri, United States of America

☯ These authors contributed equally to this work.
¤a Current address: Department of Biochemistry & Molecular Genetics, University of Colorado, Denver, United States of America
¤b Current address: Department of Molecular and Cellular Biology, Harvard University, Cambridge, United States of America
* sez@stowers.org

**Data Availability Statement:** All relevant data are within the manuscript and its Supporting Information Files. Additional raw data and images

## Abstract

Meiotic drivers bias gametogenesis to ensure their transmission into more than half the off-spring of a heterozygote. In *Schizosaccharomyces pombe*, *wtf* meiotic drivers destroy the meiotic products (spores) that do not inherit the driver from a heterozygote, thereby reducing fertility. *wtf* drivers encode both a Wtf[poison] protein and a Wtf[antidote] protein using alternative transcriptional start sites. Here, we analyze how the expression and localization of the Wtf proteins are regulated to achieve drive. We show that transcriptional timing and selective protein exclusion from developing spores ensure that all spores are exposed to Wtf4[poison], but only the spores that inherit *wtf4* receive a dose of Wtf4[antidote] sufficient for survival. In addition, we show that the Mei4 transcription factor, a master regulator of meiosis, controls the expression of the *wtf4[poison]* transcript. This transcriptional regulation, which includes the use of a critical meiotic transcription factor, likely complicates the universal suppression of *wtf* genes without concomitantly disrupting spore viability. We propose that these features contribute to the evolutionary success of the *wtf* drivers.

## Author summary

Genomes are often considered a collection of 'good' genes that provide beneficial functions for the organism. From this perspective, disease is thought to arise due to disfunction of 'good' genes. For example, infertility can be caused by the failure of a gene that normally helps fertility. This view is incomplete as 'parasitic' genes that provide no benefit to the organism also exist. These genes can also contribute to disease, often as a result of the mechanisms they use to ensure their transmission to the next generation. For example, killer meiotic drivers are found throughout eukaryotes and contribute to infertility by

can be found at the Stowers Original Data Repository: ftp://odr.stowers.org/LIBPB-1562. Instructions for accessing data from the Stowers original data repository via FTP can be found at https://www.stowers.org/research/publications/odr#ftp.

**Funding:** This work was supported by The Stowers Institute for Medical Research (SEZ); National Institutes of Health (NIH) R00GM114436 and DP2GM132936 (SEZ); the Searle Scholars program (SEZ); National Cancer Institute of the NIH under award number F99CA234523 (MABN); and the Eunice Kennedy Shriver National Institute of Child Health & Human Development of the NIH under Award Number F31HD097974 (NLN). The funders had no role in study design, data collection and analysis, or manuscript preparation. The content is solely the responsibility of the authors and does not necessarily represent the official views of the funders.

**Competing interests:** I have read the journal's policy and the authors of this manuscript have the following competing interests: NLN, MABN, SEZ: Inventor on patent application based on wtf meiotic drivers. Patent application serial 62/491,107. The other authors declare that no competing interests exist.

actively destroying the gametes (e.g., egg and sperm) that do not inherit them. In this work we study the transcriptional regulation of *wtf4*, a model killer meiotic driver found in fission yeast to understand mechanisms of drive. The *wtf4* gene encodes both a poison and an antidote protein on largely overlapping coding sequences. We found that different promoters and differential localization properties of the poison and antidote proteins both facilitate killer meiotic drive. We also found that the expression of the poison protein relies on a key transcription factor essential for gametogenesis. The use of this transcription factor likely complicates suppressing *wtf4* without compromising gametogenesis. This feature likely contributes to the evolutionary success of the *wtf* drivers, which are found in many copies in fission yeast genomes.

## Introduction

The transmission of most eukaryotic genes follows Mendel's first law of segregation. This law stipulates that the two alleles of a heterozygous organism (e.g., A/a) segregate randomly into gametes such that each allele is transmitted to 50% of the progeny [1]. There are, however, alleles that can break Mendel's law to force their own transmission into more than half of the offspring. These lawbreaking genes are called meiotic drivers [2,3]. There is a tremendous diversity of meiotic drive genes with distinct evolutionary origins and mechanisms found throughout eukaryotes [4–28]. However, the molecular details underlying how these systems are expressed and function are limited. Uncovering these details is important for understanding meiotic drive and, more broadly, has the potential to reveal novel insights about gametogenesis. For example, the existence of a sperm-autonomous phenotype, despite cytoplasmic connections between sister sperm, was discovered through study of the *t*-haplotype driver in mouse [14].

Meiotic drivers can generally be considered selfish or parasitic genes [29]. This is because drivers can persist in genomes due to the transmission advantages of drive, rather than due to fitness benefits they provide to the organisms that carry them. In fact, meiotic drivers often cause decreased fitness through a variety of direct and indirect mechanisms [30–33]. The fitness costs are especially deleterious amongst the class of drivers known as the killer meiotic drivers (reviewed in [34]). These drivers can achieve up to 100% transmission to viable gametes by destroying the products of meiosis that do not inherit the driver from a heterozygote.

Despite the fitness costs of killer meiotic drivers, many populations harbor these genes (reviewed in [34]). However, the fission yeast *S. pombe* may represent an extreme case. Different natural isolates of *S. pombe* contain between 4–14 predicted killer meiotic drivers from the *wtf* (with transposon fission yeast) gene family [35–38]. These genes kill the meiotic products (spores) that do not inherit them from a heterozygote using two proteins produced from two transcripts with largely overlapping coding sequences: a Wtf$^{poison}$ protein that kills spores and a Wtf$^{antidote}$ protein that rescues spores that inherit the driver (Fig 1A and 1B; [37,38]). After meiosis in *wtf* driver heterozygotes, the Wtf$^{poison}$ is found in all spores, while the Wtf$^{antidote}$ is enriched in those that inherit the *wtf* driver [37,39]. These distribution patterns explain how the driver has the potential to kill all spores, yet specifically rescues those that inherit the driver (Fig 1B). How this dual Wtf protein localization pattern is established and maintained is unclear.

Uncovering the mechanisms of meiotic drivers is important for understanding how these parasites impact the organisms that carry them. The *wtf* genes, in particular, confer no known fitness benefits and are largely responsible for the infertility observed in heterozygous *S. pombe*

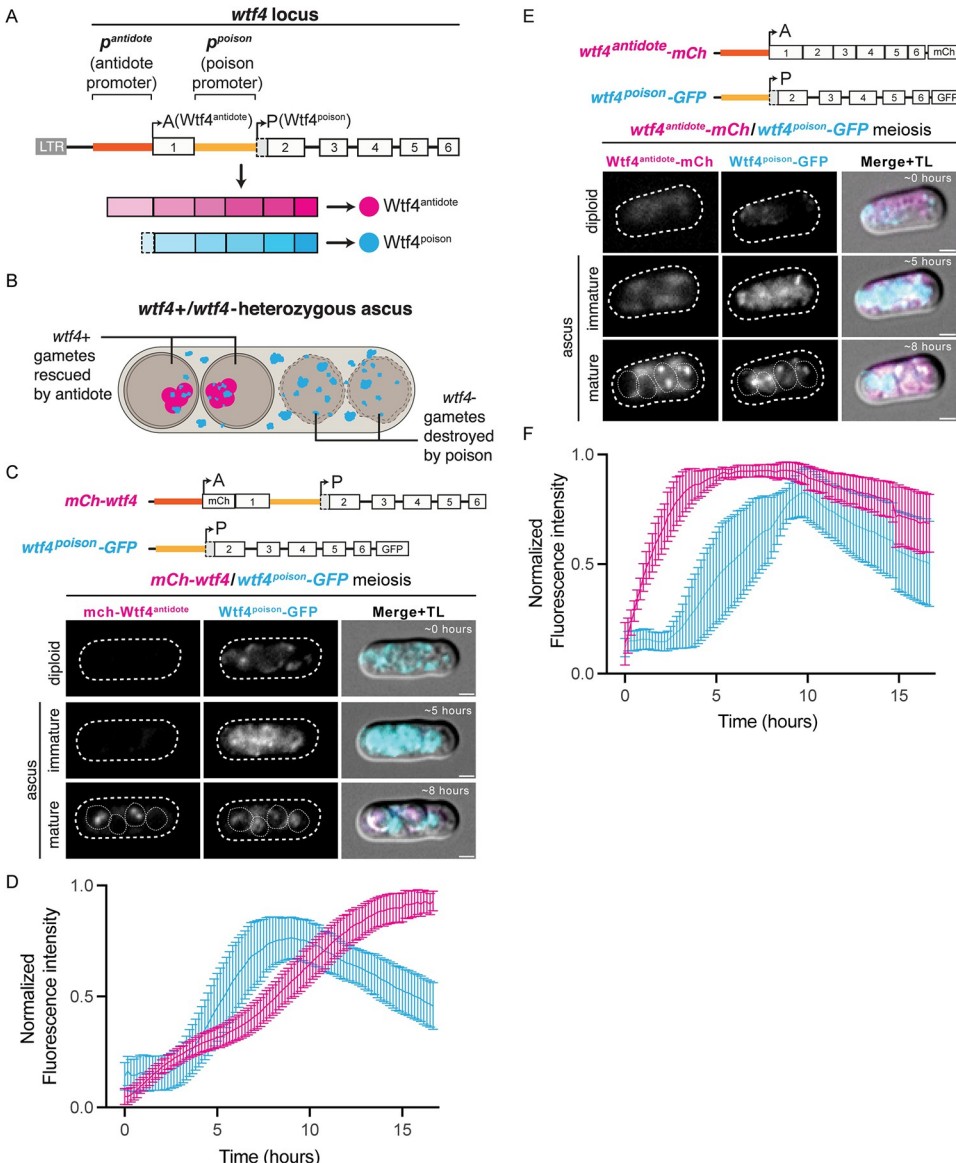

**Fig 1. Wtf4[poison] and Wtf4[antidote] proteins have distinct expression profiles.** (**A**) A depiction of the *wtf4* gene is shown. There is a Tf retrotransposon LTR located 284 base pairs upstream of exon 1. The sequence upstream of exon 1 contains the antidote promoter, $p^{antidote}$ (in orange) and intron 1 contains the poison promoter, $p^{poison}$ (in yellow). The arrows represent the predicted translational start sites. Wtf4[antidote] (magenta circle) is encoded by exons 1–6. Wtf4[poison] (cyan circle) is encoded by exons 2–6. (**B**) Model of Wtf4 poison-antidote meiotic drive from *wtf4* heterozygotes. Wtf4[poison] is present in all four spores, while Wtf4[antidote] expression is enriched in only two of the four spores. (**C**) Time-lapse microscopy of an *mCherry-wtf4/wtf4[poison]-GFP* diploid undergoing meiosis and sporulation. mCherry-Wtf4 is shown in magenta and Wtf4[poison]-GFP is shown in cyan in merged images. Images of a representative cell from the time lapse shown at ~ 0 hours (diploid), ~5 hours (immature ascus), and ~8 hours (mature ascus) after the video begins. Distinct images for these strains were first presented in [37] and a timelapse following protein movement in mature asci of these strains was presented in [52]. (**D**) Normalized fluorescence intensity plots of Wtf4[poison]-GFP (cyan line) and mCherry-Wtf4[antidote] (magenta line) over the course of the time lapse (n = 25). (**E**) Time-lapse microscopy of a *wtf4[antidote]-mCherry/wtf4[poison]-GFP* diploid undergoing meiosis and sporulation. Wtf4[antidote]-mCherry is shown in magenta and Wtf4[poison]-GFP is shown in cyan in merged images. Images of a representative cell from the time lapse shown at ~ 0 hours (diploid), ~5 hours (immature ascus), and ~8 hours (mature ascus) after the video begins. (**F**) Normalized fluorescence intensity plots of Wtf4[poison]-GFP (cyan line) and Wtf4[antidote]-mCherry (magenta line) over the course of the time lapse (n = 13) TL = transmitted light. All scale bars represent 2 μm. Error bars depict the 95% confidence interval. Time lapse imaging was performed under the same settings. Not all images are shown at the same brightness and contrast to avoid over saturation of pixels in the brighter images.

diploids generated by outcrossing between distinct isolates [40–45]. Due to the fitness costs of *wtf* drivers, *S. pombe* has likely evolved ways to tolerate or suppress drive. There are currently two described mechanisms that can suppress *wtf* meiotic drivers: suppression by other *wtf* genes and gamete disomy [39–41]. Transcriptional silencing operating through trans-acting regulators or cis-acting chromatin packaging are additional candidates for suppression mechanisms. While this possibility has not been explored during spore formation, histone deacetylation and TOR-mediated RNA processing have been shown to contribute to mitotic silencing of *wtf* genes [46–49].

Our goals in this study were to understand how the expression and localization of Wtf driver proteins are coordinated and to find possible routes to transcriptional suppression of *wtf* drivers. To do this, we use the *wtf4* gene from the *S. kambucha* natural isolate of *S. pombe* as a model system. We define the promoter sequences that control the *wtf4^poison^* and *wtf4^antidote^* transcripts. We demonstrate that differential expression due to the distinct promoters and differential inclusion of the two Wtf4 proteins in developing spores both contribute to Wtf4 protein localization and thus efficient drive. We also found that the expression of the *wtf4^poison^* transcript is controlled by the transcription factor Mei4, a master regulator of gene expression in meiosis [50,51]. In the context of our results, we discuss why evolving or maintaining transcriptional silencing of *wtf* genes and drivers in general may be challenging.

## Results

### Wtf4^poison^ and Wtf4^antidote^ proteins have distinct expression profiles

In previous work, we used fluorescent markers to separately tag and visualize the Wtf4 proteins [37,52]. We imaged mostly mature asci (sacks holding spores) and diploid cells present at the same time points that had not completed spore formation. In diploid cells, these analyses revealed signal from a tagged Wtf4^poison^-GFP allele, but not an mCherry-Wtf4^antidote^ allele. In mature asci, we observed Wtf4^poison^-GFP in all spores and mCherry-Wtf4^antidote^ strongly enriched in spores that inherited the locus encoding the allele [37,52]. These analyses, however, were too coarse to reconstruct the full expression dynamics of the two proteins.

In this work, we explored the expression of the two Wtf4 proteins more thoroughly to uncover how the two proteins achieve targeted destruction of spores that do not inherit *wtf4* from a heterozygote (Fig 1B). We again used *wtf4^poison^-GFP*, a separation-of-function allele that does not express Wtf4^antidote^ and adds a GFP tag to the C-terminus of the Wtf4^poison^ (S1 Fig allele 1). We have previously shown that this allele acts as a functional poison but is slightly less toxic than the corresponding untagged *wtf4^poison^* allele ([37], S2A Fig, diploid 4). We also used the *mCherry-wtf4* allele, which tags Wtf4^antidote^, but still generates an untagged Wtf4^poison^ (S1 Fig allele 2). This allele functions similarly to the wild type *wtf4* in allele transmission and viable spore yield (VSY) assays ([37], S2A Fig, diploid 3). Viable spore yield is a measure of fertility that assays the number of viable spores generated per cell induced to form spores [53]. In this work, we normalized viable spore yields to those measured in wild-type control cells with an empty vector integrated in the genome instead of a *wtf4* allele. Those wild-type cells are therefore considered to have 100% fertility. Complete drive in a heterozygote is expected to reduce fertility by roughly 50% and expression of a Wtf4^poison^ allele in the absence of Wtf4^antidote^ is expected to reduce fertility more than 50%.

We integrated the *mCherry-wtf4* and *wtf4^poison^-GFP* alleles mentioned above into the *ade6* locus of different haploid strains. We crossed these two strains together to generate heterozygous *mCherry-wtf4 / wtf4^poison^-GFP* diploids. We then completed time-lapse microscopy during fission yeast meiosis and spore formation (Fig 1C and 1D). In this text, we depict Wtf4^antidote^ in magenta and Wtf4^poison^ in cyan in all the images, regardless of fluorescent

protein tag. We consistently observed Wtf4$^{poison}$-GFP hours before mCherry-Wtf4$^{antidote}$ (Fig 1C and 1D). Given that GFP maturation is only minutes faster than mCherry [54–56], the earlier signal from Wtf4$^{poison}$-GFP cannot be explained by maturation times of the fluorescent proteins alone. Through the time-course, the GFP signal decreases, but then increases to reach high intensity prior to spore formation. The signal was found in all four spores, not just those that inherited the *wtf4$^{poison}$-GFP* allele (Fig 1C and 1D, S1 Video panel A). These observations suggested either that most of the Wtf4$^{poison}$ protein is produced prior to spore individualization or that the *wtf4$^{poison}$* transcript and/or protein is freely exchanged between spores. We distinguished between these models by photobleaching two spores or the entire ascus in asci expressing Wtf4$^{poison}$-GFP and assaying recovery of fluorescence (i.e., FRAP). We found minimal recovery of Wtf4$^{poison}$-GFP signal after photobleaching both spores and asci (S3A–S3D Fig). These data support that most Wtf4$^{poison}$ is produced prior to spore formation and then packaged within all spores.

The mCherry-Wtf4$^{antidote}$ signal was very low prior to spore formation and reached max intensity after spore formation (Fig 1C and 1D, S1 Video panel A). We were curious if the transcription of the *wtf4$^{poison}$* could interfere with the transcription of the *wtf4$^{antidote}$* prior to spore formation because the *wtf4$^{poison}$* transcriptional start site is downstream of the *wtf4$^{antidote}$* transcriptional start site. To test this, we generated an alternate allele of *mCherry-wtf4$^{antidote}$* that lacks introns, and thus lacks the capacity to produce a poison transcript (Figs 1A and S1 allele 3). This allele encodes a functional antidote as it suppresses drive of wild type *wtf4*, but does not cause drive alone (S2A Fig, diploids 10 and 11). In heterozygotes with this allele integrated at *ade6*, we again saw minimal signal from this mCherry-Wtf4$^{antidote}$ protein in diploids and strong enrichment in two spores. This suggests *wtf4$^{poison}$* expression is not significantly affecting the gross localization dynamics of mCherry-Wtf4$^{antidote}$ (S2B Fig).

We also tested if the difference in signal we observed between the tagged Wtf4$^{antidote}$ and Wtf4$^{poison}$ proteins could be due to differences in the N- versus C-terminal tags. To test this, we assayed a C-terminally tagged Wtf4$^{antidote}$ separation-of-function allele, *wtf4$^{antidote}$-mCherry*. This allele does not encode Wtf4$^{poison}$ because it lacks introns (Figs 1A and S1 allele 4). We integrated this allele into the *ura4* locus and found that it indeed acted as an antidote-only allele because it does not drive on its own, but it does suppress drive of *wtf4* (S2A Fig, diploids 8 and 9). We then crossed a strain carrying the *wtf4$^{antidote}$-mCherry* allele to haploid cells carrying *wtf4$^{poison}$-GFP* to generate *wtf4$^{antidote}$-mCherry/ura4+, wtf4$^{poison}$-GFP/ade6+* diploids. We imaged these diploids over a time course and observed that the Wtf4$^{antidote}$ localization was different than the pattern we observed with the *mCherry-wtf4* allele in that the cells had mCherry signal prior to spore formation (Fig 1E and 1F, S1 Video panel B). Specifically, we saw Wtf4$^{antidote}$-mCherry in diploid cells induced to undergo meiosis and in the ascal cytoplasm (i.e., outside of spores) of immature asci (Figs 1E and 1F and S2C, S1 Video panel B). In mature asci, Wtf4$^{antidote}$-mCherry was strongly enriched in two of the four spores, similar to the N-terminally tagged protein, but significant signal remained in the ascal cytoplasm outside spores Figs 1E and 1F and S2C, S1 Video panel B). Another C-terminally tagged Wtf4$^{antidote}$ separation-of-function allele, *wtf4$^{antidote}$-GFP* (S1 Fig allele 5), localized similarly to *wtf4$^{antidote}$-mCherry* (S2C and S2D Fig) and behaved as a functional antidote allele (S2A Fig, diploids 13 and 15).

We assayed expression of the Wtf4 proteins via western blots. In samples from *wtf4$^{antidote}$-mCherry/ura4+, wtf4$^{poison}$-GFP/ade6* diploids, we detected free GFP and a band we infer to be full-length Wtf4$^{poison}$-GFP when cells are undergoing meiosis and when mature spores are present. We never detect full-length Wtf4$^{antidote}$-mCherry, but we detect free mCherry when cells are undergoing meiosis and when mature spores are present (S4 Fig). In diploids expressing the *mcherry-wtf4$^{antidote}$* allele, we also only detect free mCherry, but only when mature

spores are present, consistent with our image analyses showing mCherry only in spores (S5 Fig). Our inability to detect full-length tagged Wtf4$^{antidote}$ on a western was not surprising as we previously showed the protein is quickly trafficked to vacuoles, where it is presumably degraded [52].

We also attempted to detect Wtf4$^{poison}$-GFP and Wtf4$^{antidote}$-mCherry on Western blots from samples collected from cells induced to undergo synchronous meiosis using an analog sensitive allele of the Pat1 kinase that inhibits meiotic entry (*pat1.L95G*; [57]). The results were similar to our results in wild-type cells, described above, except we could detect a faint band early in meiosis that may be Wtf4$^{antidote}$-mCherry, in addition to the free mCherry band observed at all timepoints (S6 Fig). Overall, the westerns support our imaging as we see signal from the tagged Wtf4$^{poison}$ locus when cells are undergoing meiosis and in samples containing spores. We observe signal from both tagged Wtf4$^{antidote}$ alleles in samples containing spores, but only from the C-terminally tagged allele in samples collected prior to spore formation (S5 and S6 Figs).

We conclude that the C-terminally tagged alleles reveal a Wtf4$^{antidote}$ protein population present prior to spore formation that is not apparent with the N-terminally tagged alleles. The nature of the additional protein is not clear. The early Wtf4$^{antidote}$ protein could be produced using the second translational start site found in exon 1 (codon 12) via leaky transcriptional scanning [58]. This second start site can be used to encode a fully functional antidote protein (S1 Fig allele 6, S2A Fig diploids 6–7, [37]). In the two N-terminally Wtf4$^{antidote}$ tagged alleles discussed above, the putative shorter Wtf4$^{antidote}$ protein would not be tagged, even if it was produced. It is also possible the Wtf4$^{antidote}$ population revealed with the C-terminal tag is full-length protein, but its expression prior to spore formation is disrupted by the N-terminal tag. We did not distinguish between these possibilities.

Overall, our results show that signal from the N-terminally tagged Wtf4$^{antidote}$ used in our previous studies does not reflect the total Wtf4$^{antidote}$ population. The C-terminally tagged alleles described here reveal additional Wtf4$^{antidote}$ protein is present prior to spore formation. This interpretation is also consistent with previous long-read RNA sequencing data showing at least some transcription of *wtf* antidotes prior to spore formation (e.g., 0–6 hours after meiotic induction) [37,38,59]. Both alleles show that additional Wtf4$^{antidote}$ protein production occurs after spore formation. In contrast, our time course analysis of Wtf4$^{poison}$-GFP production revealed that most of the Wtf4$^{poison}$ protein is present prior to spore formation and the protein cannot move freely between spores.

## Distinct promoters contribute to the distinct localization patterns of the Wtf4 proteins

We next wanted to explore if the distinct promoters contribute to the different localization patterns of the Wtf4$^{poison}$ and Wtf4$^{antidote}$ proteins in asci. To test this, we generated constructs with fluorescent proteins, mCherry and GFP, under the control of the *wtf4$^{antidote}$* and *wtf4$^{poison}$* promoters (including the 5' untranslated regions (UTRs)), respectively. For the *wtf4$^{antidote}$* promoter (*p$^{antidote}$*), we used the 285 base pairs found upstream of exon 1. This is just downstream of a nearby Tf transposon long terminal repeat (LTR) that is not necessary for Wtf$^{antidote}$ production (Fig 1A; [38,52]). Previous work assaying *cw9* and *cw27* (two *wtf* drivers in the CBS5557 isolate) found that the 288 bp upstream of exon 1 was insufficient to generate a fully functional *p$^{antidote}$* promoter [38]. Although the promoter regions of the genes are highly similar, our previous results showed that for *wtf4*, 285 bp upstream sequence was sufficient to promote production of a functional Wtf4$^{antidote}$ [52]. This sequence is also well conserved amongst *wtf* genes that encode for an antidote, which also supports that this region includes

the promoter of $wtf4^{antidote}$ ([36–38,59,60]; Fig 2A). To characterize the poison promoter ($p^{poison}$), we used the 230 bp sequence that makes up intron 1 of the antidote transcript (Fig 2D). This sequence includes the transcriptional start site of the $wtf4^{poison}$ transcript and is well conserved amongst $wtf$ drivers (Fig 2D, [36–38,59]).

We integrated the promoter reporter constructs at the $ade6$ locus of different haploid strains. We then generated diploid cells heterozygous for each of the reporters individually (i.e., $reporter/ade6+$) and imaged them through meiotic induction and spore formation. With the $p^{antidote}$-$mCherry$ reporter, we observed low signal in diploids undergoing meiosis, but the strongest mCherry signal was observed in two out of the four spores (Fig 2B and 2C), presumably the two that inherited the reporter construct. Importantly, the observed localization of the $p^{antidote}$-$mCherry$ reporter protein supports that expression occurs both before and after spore individualization.

As mentioned above, a fully functional Wtf4$^{antidote}$ protein can be made using a second ATG codon at the 12$^{th}$ codon position [37]. We speculated that additional transcriptional regulatory sequences may be found in those coding sequences upstream of codon 12. To test this, we generated $p^{antidote\ long}$-$mCherry$, a construct with mCherry under a $wtf4^{antidote}$ promoter that also contains the first 11 codons of the $wtf4^{antidote}$ coding sequence (S7B Fig). We integrated this allele at $lys4$ and crossed this strain to wild type to generate heterozygous diploids. In these diploids and in the asci generated via these diploids, we saw mCherry expressed from the $p^{antidote\ long}$-$mCherry$ reporter at a similar level to the mCherry expressed from the shorter $p^{antidote}$ (S7A–S7C Fig). The simplest interpretation of our data is that constructs under the $p^{antidote}$ promoter are expressed before and after spore formation. In addition, the pattern of reporter signal enrichment we observe suggests the transcripts and proteins produced in a spore can be retained within that spore. We did not directly explore a role of the 5' UTR in ensuring spore-specific expression, but it is possible the UTR could affect translation or transcript retention in the spore.

For the $p^{poison}$-$GFP$ reporter, we observed expression in diploid cells induced to undergo meiosis. We also observed similar GFP reporter signal among the four spores (Fig 2E and 2F). These observations are similar to the localization patterns we observe with Wtf4$^{poison}$-GFP (e.g., Fig 1C). This roughly equal distribution of $p^{poison}$-$GFP$ reporter signal in the four spores, even though only two inherited the $p^{poison}$-$GFP$ reporter, was starkly different from the two-spore enrichment observed with $p^{antidote}$-$mCherry$ reporter. Assuming equal spore to spore mobility between GFP and mCherry, this further supports that the majority of the the $p^{poison}$ transcripts are produced prior to spore individualization. As with the $p^{antidote}$, it is possible that the 5' UTR affects the translation or mobility of transcripts driven by $p^{poison}$ promoter, but we did not explore these possibilities.

Together, these experiments demonstrate differential gene expression patterns under the $p^{antidote}$ and $p^{poison}$ promoters. Our results also show that these different promoters largely contribute to the localization of Wtf4$^{poison}$ protein within all spores and the enrichment of the Wtf4$^{antidote}$ protein within the spores that inherit the $wtf4$ locus.

## Master meiotic regulator, Mei4, controls Wtf4$^{poison}$ expression

We next sought to explore transcriptional regulation of the $wtf4$ transcripts. Previous work noted that the transcription of multiple $wtf$ genes was controlled by the fork-head transcription factor Mei4, as $wtf$ transcription is decreased if $mei4$ is deleted and $wtf$ gene expression is increased if Mei4 is overexpressed [61]. Mei4 controls the expression of hundreds of genes and is known as the master regulator of middle meiosis genes [50,51]. However, the prior study was performed before the discovery that $wtf$ drivers can make two transcripts, so it was not clear which transcript Mei4 controls.

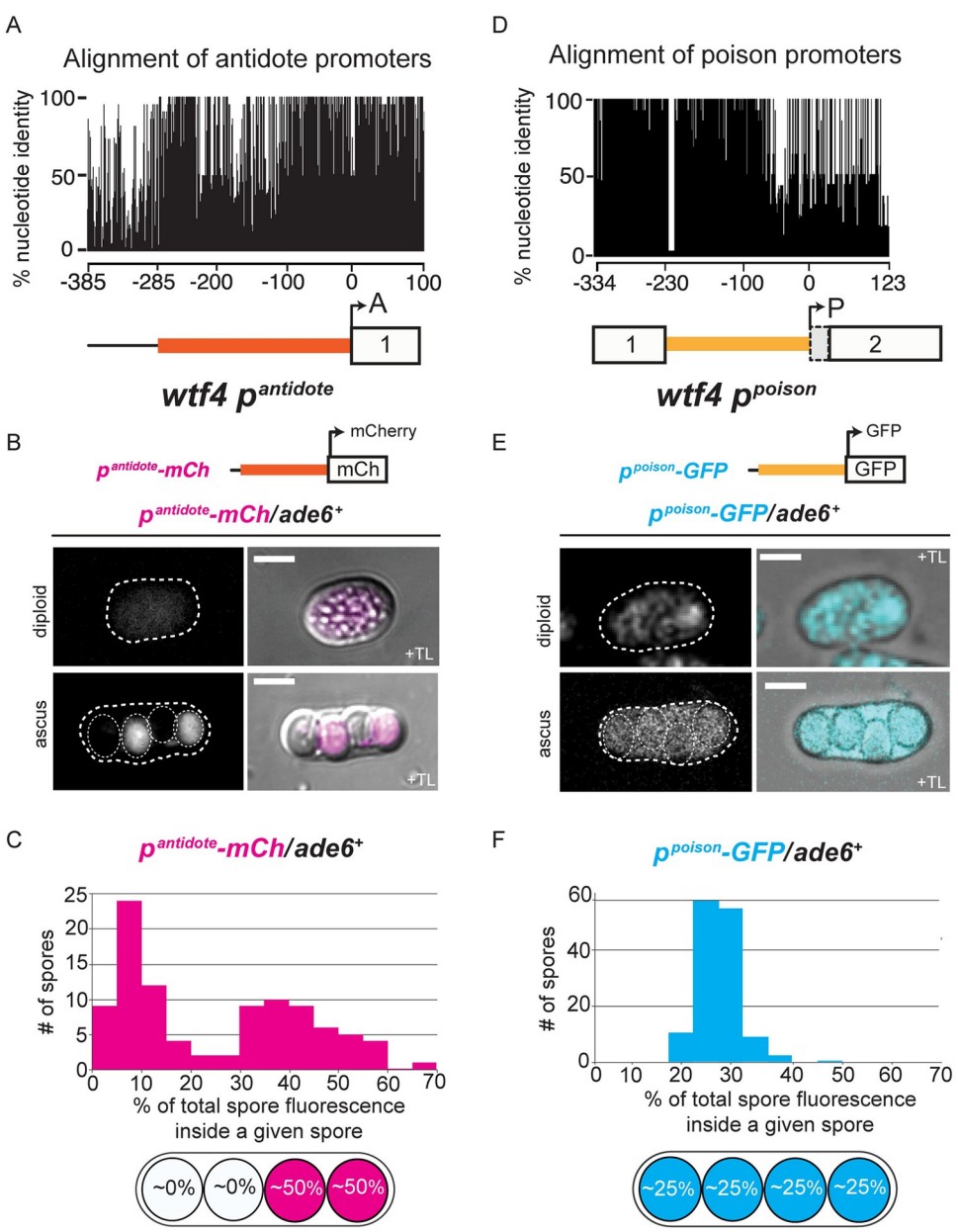

**Fig 2. Distinct promoters largely explain differential localization of Wtf4$^{poison}$ and Wtf4$^{antidote}$ proteins.**
Depictions and alignments of (**A**) the *wtf4* antidote promoter and (**D**) the *wtf4* poison promoter are shown. For the alignment of antidote promoters, we aligned the 600 base-pairs upstream of 41 predicted antidote-only alleles [36] from three different strains of *S. pombe* (the reference genome, *S. kambucha*, and *FY29033*, [85]). For the alignment of poison promoters, we aligned the intron 1 sequences (flanked by sequences 100 base-pairs upstream and downstream of intron 1) of 28 predicted poison-antidote *wtf* drivers [36] from three different strains of *S. pombe* (the reference genome, *S. kambucha*, and *FY29033*). The image shows the percent identity at each nucleotide position, excluding gaps. (**B**) Images of $p^{antidote}$-mCherry/ade6+ diploid and ascus. (**C**) Quantification of mCherry fluorescence within $p^{antidote}$-mCherry/ade6+ asci (n = 24). (**E**) Images of $p^{poison}$-GFP/ade6+ diploid and ascus. (**F**) Quantification of GFP fluorescence within $p^{poison}$-GFP/ade6+ asci (n = 34). All images were acquired after 3 days on sporulation media. For quantification, we assayed the fluorescence intensity within each spore and then divided that number by the total fluorescence intensity within all 4 spores. TL = transmitted light. All scale bars represent 2 μm. Images of diploids are shown at the same brightness and contrast and were imaged at the same settings as the ascus generated from diploids of the same genotype.

To understand which *wtf* transcript Mei4 controls, we first looked for the Mei4-binding motif in the *wtf4* promoters. Fork-head transcription factors bind sequences containing FLEX motifs and the complete Mei4 binding motif contains a nine-base pair (GTAAACAAA) core sequence [50,51,62,63]. We found this nine-base pair Mei4 binding motif in the $p^{poison}$ promoter 110 base pairs upstream of the $wtf4^{poison}$ translational start site (Fig 3A). Moreover, we found this sequence was conserved amongst known *wtf* drivers in *S. pombe* (Fig 3A) and in *wtf* drivers found in other *Schizosaccharomyces* species [64]. In contrast, the $p^{antidote}$ promoter does not contain a FLEX motif.

To test if Mei4 binds the $p^{poison}$ FLEX motif in meiotic cells, we analyzed data from a previous Chromatin-Immunoprecipitation sequencing (ChIP-seq) experiment of Mei4 during meiosis done in the lab isolate of *S. pombe* [50]. Due to the similarity of the *wtf* genes, many reads could not be uniquely assigned to *wtf4* or any single *wtf* gene and were thus randomly assigned to matching sites. The sequence of intron 1, however, is distinct between *wtf* drivers that contain the $p^{poison}$ promoter and antidote-only *wtf* genes that do not contain the $p^{poison}$ promoter, allowing us to distinguish Mei4 binding between intact *wtf* drivers and other *wtf* genes [36,38,39]. Prior to meiosis, we observed that Mei4 binding to *wtf* drivers (*wtf4*, *wtf13*) was low relative to the genome average (Fig 3B, 0 hours). After 4 hours in meiosis, when most cells are in prophase I, we saw a strong increase in reads in the $p^{poison}$ promoter (intron 1) of the *wtf* drivers (Fig 3B, 4 hours). We did not see an increase in Mei4 binding to antidote-only *wtf* genes (*wtf5*, *wtf9*, *wtf10*, *wtf16*, *wtf18*, *wtf20*, *wtf21*, and *wtf25*) during meiosis (Fig 3B). We also saw a Mei4 peak at the C-terminal region of both subsets of *wtf* genes. We predict this peak is due to the nearby LTRs, as we saw high Mei4 binding on LTRs independently of their association with *wtfs* (S8 Fig).

To further test the idea that $wtf4^{poison}$ expression is controlled by Mei4, we compared two tagged alleles: a fully functional allele (*wtf4-GFP*, [37]) and an allele lacking the FLEX motif (*wtf4$^{FLEXΔ}$-GFP*; S1 Fig alleles 7 and 8). We integrated the alleles at *ade6* to test function. We did not anticipate that deletion of the Mei4-binding motif would affect the expression or function of the Wtf4$^{antidote}$ protein as we can delete all of intron 1 from *wtf4* (as in *wtf4$^{antidote}$-mCherry*), without affecting Wtf4$^{antidote}$ function (S1 Fig allele 4, S2A Fig diploids 8 and 9). Consistent with our expectations, the *wtf4$^{FLEXΔ}$-GFP* allele encodes a functional antidote as it suppresses drive of wild-type *wtf4* in a *wtf4$^{FLEXΔ}$-GFP /wtf4+* heterozygote (Fig 3C, diploid 19). In addition, we saw strong GFP signal within two of the four spores in asci generated by *wtf4$^{FLEXΔ}$-GFP/ade6+* diploids (Fig 3D), similar to tagged Wtf4$^{antidote}$ alleles (e.g., S2C and S2D Fig). We tested the ability of the *wtf4$^{FLEXΔ}$-GFP* allele to encode a functional Wtf4$^{poison}$ by assaying if the allele could drive in a *wtf4$^{FLEXΔ}$-GFP/ade6+* heterozygote. These diploids had Mendelian allele transmission (Fig 3C, diploid 18). As the *wtf4$^{FLEXΔ}$-GFP* does encode a functional antidote, we conclude that the allele does not drive because the Mei4-binding motif is essential for production of the Wtf4$^{poison}$. These data, combined with the previous genetic and biochemical data of others [50,61] support that Mei4 controls expression of the Wtf4$^{poison}$ and likely the poison proteins produced by other *wtf* drivers.

## Distinct transcriptional regulation of Wtf4$^{poison}$ and Wtf4$^{antidote}$ promote efficient meiotic drive

To explore the functional relevance of the differential transcriptional of $wtf4^{poison}$ and $wtf4^{antidote}$, we generated alleles that expressed the antidote protein from the $p^{poison}$ promoter and *vice versa*. We used the $p^{poison}$-*wtf4$^{antidote}$-GFP* allele (integrated at *ade6*), to test if the pattern of $p^{antidote}$ transcription was important for drive (S1 Fig allele 9). We compared the phenotype of this promoter swap allele to the functional *wtf4$^{antidote}$-GFP* allele (integrated at *ade6*) with the

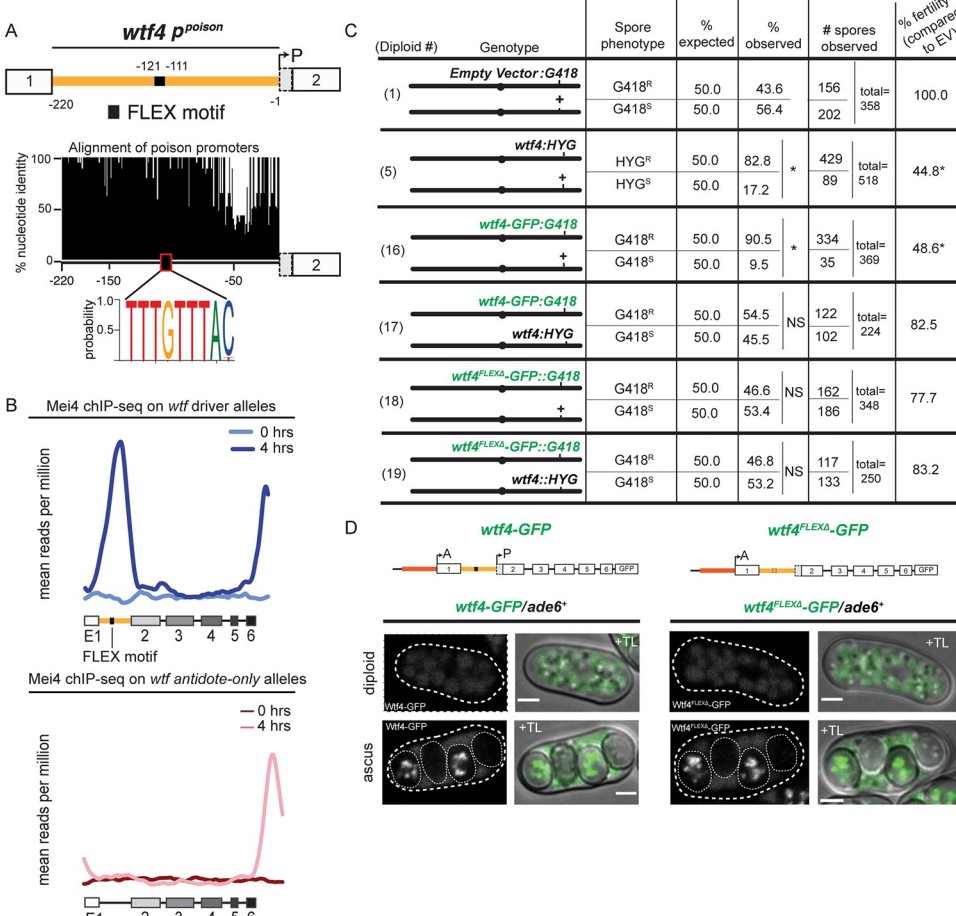

**Fig 3. Master meiotic regulator, Mei4, controls Wtf4^poison transcription.** (**A**) Depiction of the *wtf4*^poison promoter contained within intron 1. There is a FLEX motif (black box, TGTTTAC in opposite orientation) located 111 to 117 nucleotides upstream of the Wtf4^poison transcriptional start site. Percent consensus nucleotide identity of intron 1 sequences of 28 predicted poison-antidote *wtf* drivers [36] from three different strains of *S. pombe* (the reference genome, *S. kambucha*, and *FY29033*). The figure shows the percent consensus identity at each nucleotide position, excluding gaps. (**B**) Mei4 Chip-seq data (data from [50]) showing Mei4 binding on *wtf* genes at both 0 hours and 4 hours after meiotic induction. On the top panel, the reads are aligned to the predicted *wtf* meiotic drive genes (*wtf4*, *wtf13*) and on the bottom the reads are aligned to the *wtf* genes that are predicted to only encode antidote proteins (*wtf5*, *wtf9*, *wtf10*, *wtf16*, *wtf18*, *wtf20*, *wtf21*, *wtf25*) in the *S. pombe* genome version ASM294v2. For any reads that mapped to more than one location, only a single location, chosen at random, is reported. (**C**) Allele transmission and fertility (assayed via viable spore yield, VSY) of diploids with the depicted genotype. Alleles listed are at *ade6*. The genotype column shows a cartoon depiction of the relevant genotype. The progeny phenotypes are then shown on the right. For diploids heterozygous at one locus (e.g., Diploid 1), two values are shown (top and bottom) that represent the two possible haploid genotypes. Spores exhibiting both parental phenotypes were considered diploid or aneuploid and were excluded from this table but can be found in S1 Data. The expected values assume Mendelian allele transmission. (* = p < 0.05, NS = not significant; G-test for allele transmission, Wilcoxon test for VSY, in comparison to the empty vector control). We compared all diploids to diploid 1 as the control. The data for diploid 1 is also depicted in S2A and S9A Figs. Diploid 5 is also shown S2A Fig. (**D**) Images of heterozygous *wtf4-GFP/ade6+* (left) and *wtf4*^FLEXΔ-*GFP/ade6+* (right) diploids and asci acquired after 1 day on sporulation media. Wtf4-GFP and Wtf4^FLEXΔ-GFP are shown in green. Scale bar represents 2 μM. TL = transmitted light. All images are shown at the same brightness and contrast for accurate comparison.

endogenous promoter described above (S1 Fig allele 5). We found that the *p*^poison-*wtf4*^antidote-*GFP* was defective in meiotic drive. Specifically, the *p*^poison-*wtf4*^antidote-*GFP* allele was unable to suppress the drive of an intact *wtf4* driver in a *p*^poison-*wtf4*^antidote-*GFP/wtf4* heterozygote (S9A Fig diploid 22). The control allele (*wtf4*^antidote-*GFP*) fully suppressed drive of intact *wtf4* (S9A

Fig diploid 15). We also tested the ability of the $p^{poison}$-$wtf4^{antidote}$-GFP to suppress the spore inviability caused by an allele that expresses only Wtf4$^{poison}$ (mCherry-$wtf4^{poison}$; S1 Fig allele 10). We found that the $p^{poison}$-$wtf4^{antidote}$-GFP provided only minimal rescue of spore death and that the rescue was restricted to the spores that inherited the $p^{poison}$-$wtf4^{antidote}$-GFP (S9A Fig compare diploids 12 and 21).

Our results led us to question why Wtf4$^{poison}$ expressed from the $p^{poison}$ promoter was able to poison all spores, but the Wtf4$^{antidote}$ expressed from the $p^{poison}$ promoter was not able to rescue all spores. Such a rescue was expected as we previously observed that Wtf4$^{poison}$ ectopically induced in mitotic cells is neutralized by expression of Wtf4$^{antidote}$ from a matching promoter [52]. To explain this discrepancy, we hypothesized that the Wtf4$^{antidote}$ may be more likely than the Wtf4$^{poison}$ to be excluded from spores as they individualize within the ascal cytoplasm. This hypothesis was based on our images of Wtf4$^{antidote}$ proteins with C-terminal tags that tended to show more signal in the ascal cytoplasm than within all newly developed spores. There also tended to be more signal in the ascal cytoplasm than within spores that did not inherit the locus encoding the Wtf4$^{antidote}$ in mature asci (Figs 1E, S2C and S2D). If the Wtf4$^{antidote}$ is excluded from spores as they individualize more than Wtf4$^{poison}$, Wtf4$^{antidote}$ would not be expected to protect spores well unless it was expressed in them.

To formally test our hypothesis, we measured the fraction of the Wtf4$^{antidote}$-GFP and mCherry-Wtf4$^{poison}$ protein found within spores in asci generated by diploids carrying the $p^{poison}$-$wtf4^{antidote}$-GFP and/or the mCherry-$wtf4^{poison}$ allele (Fig 4A–4D). We found that a larger fraction of the mCherry-Wtf4$^{poison}$ signal was in spores, relative to the fraction of total Wtf4$^{antidote}$-GFP signal found in spores (Fig 4D, left panel). This suggests that the Wtf4$^{antidote}$ protein is excluded from spores more than the Wtf$^{poison}$ as spores individualize. In addition, we found that a smaller fraction of the mCherry-Wtf4$^{poison}$ signal was in spores produced by diploids carrying the $p^{poison}$-$wtf4^{antidote}$-GFP allele, compared to diploids not expressing a $wtf4^{antidote}$ allele. This suggests the Wtf4$^{antidote}$ can promote the exclusion of some Wtf4$^{poison}$ from spores as they individualize.

To test if the $p^{poison}$ transcriptional timing was important for drive, we generated an allele with Wtf4$^{poison}$-GFP expressed by the $p^{antidote}$ promoter ($p^{antidote}$-$wtf4^{poison}$-GFP; S1 Fig allele 11). However, we failed to transform this construct into cells not carrying a $wtf4^{antidote}$ allele, despite multiple attempts and using multiple transformation protocols (standard lithium acetate and electroporation; [65,66]). We were also unable to generate strains carrying $p^{antidote}$-$wtf4^{poison}$-GFP in the absence of $wtf4^{antidote}$ via mutation or genetic crosses (S9A Fig diploid 24). These results suggest that the $p^{antidote}$ promoter may cause some Wtf4$^{poison}$ expression during vegetative growth, killing cells lacking $wtf4^{antidote}$. Consistent with this, published long-read sequencing data shows some transcription of $wtf^{antidote}$ alleles occurs in cells prior to meiotic induction [59].

We assayed the ability of the $p^{antidote}$-$wtf4^{poison}$-GFP allele (at ade6) to support drive in cells heterozygous for $wtf4^{antidote}$-mCherry (at ura4). We compared the $p^{antidote}$-$wtf4^{poison}$-GFP allele to a control mCherry-$wtf4^{poison}$ allele. We found that the $p^{antidote}$-$wtf4^{poison}$-GFP allele supported minimal drive of the $wtf4^{antidote}$-mCherry allele, while the control mCherry-$wtf4^{poison}$ allele supported strong drive of $wtf4^{antidote}$-mCherry (S9A Fig, diploids 23 and 24). Importantly most of the drive of the $wtf4^{antidote}$-mCherry allele induced by $p^{antidote}$-$wtf4^{poison}$-GFP stems from the poison allele killing spores that inherit it (100% drive in spores that inherit $p^{antidote}$-$wtf4^{poison}$-GFP), with minimal killing of spores that do not (58% drive in spores that do not inherit $p^{antidote}$-$wtf4^{poison}$-GFP). In the control cross, drive of the $wtf4^{antidote}$-mCherry allele was 100% and 91% in the spores that did and did not inherit the mCherry-$wtf4^{poison}$ allele, respectively (S9A Fig, diploids 23 and 24).

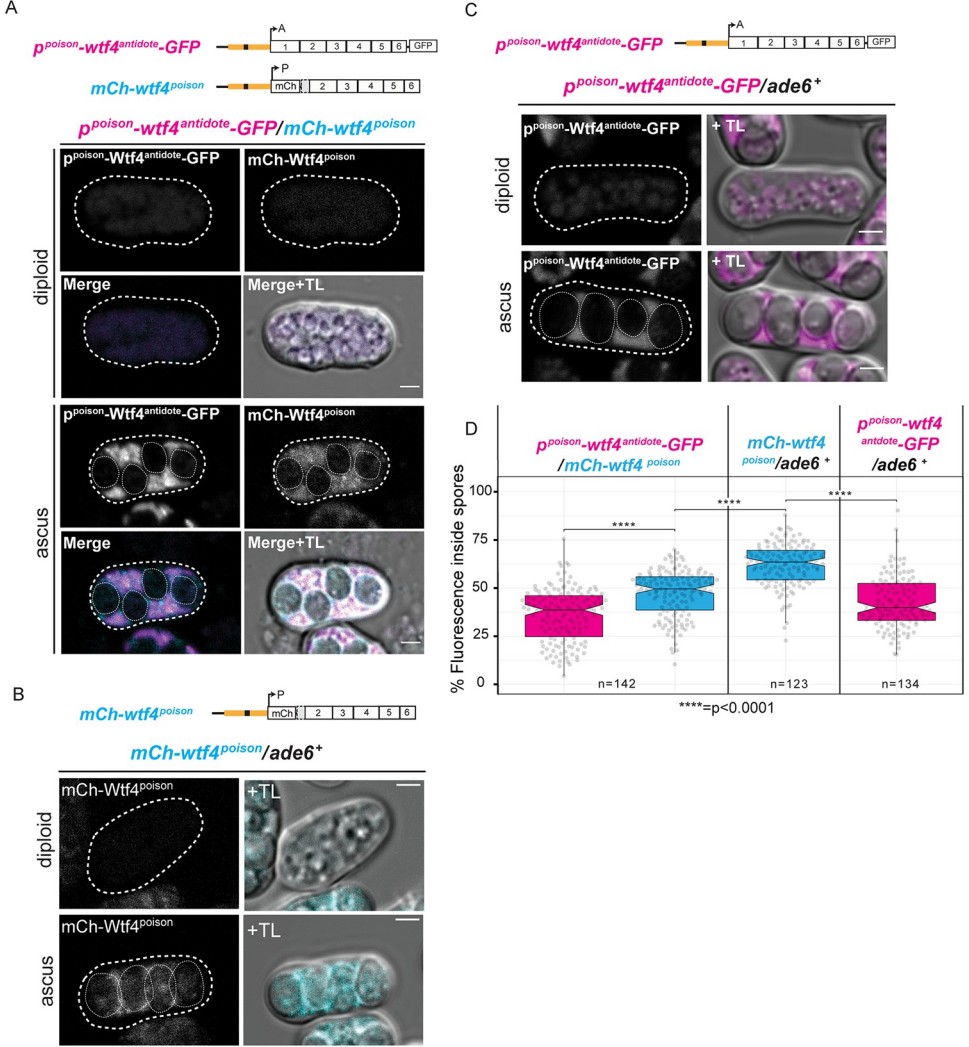

**Fig 4. Wtf4$^{antidote}$ is excluded from developing spores more than the Wtf4$^{poison}$.** (**A**) Image of a $p^{poison}$-wtf4$^{antidote}$-GFP/ mCherry-wtf4$^{poison}$ diploid and ascus. (**B**) Image of a mCherry-wtf4$^{poison}$/ ade6+ diploid and ascus. (**C**) Image of $p^{poison}$-wtf4$^{antidote}$-GFP/ ade6+ diploid and ascus. (**D**) Quantification of the percentage of the total fluorescence intensity found inside of spores of mature asci produced by $p^{poison}$-wtf4$^{antidote}$-GFP/ mCherry-wtf4$^{poison}$ (left, n = 142), mCherry-wtf4$^{poison}$/ade6$^+$ diploids (center, n = 123) and $p^{poison}$-wtf4$^{antidote}$-GFP/ade6$^+$ diploids (right, n = 134). $p^{poison}$-Wtf4$^{antidote}$-GFP fluorescence data is depicted in magenta while mCherry-Wtf4$^{poison}$ is depicted in cyan. Error bars depict the 95% confidence interval. $p^{poison}$-Wtf4$^{antidote}$-GFP is shown in magenta and mCherry-Wtf4$^{poison}$ is shown in cyan in merged images. TL = transmitted light. All scale bars represent 2 μm. All images were acquired after 1 day on sporulation media. Images were taken at the same settings and are shown at the same brightness and contrast for accurate comparison.

Our genetic results indicate that the Wtf4$^{poison}$ expressed from the $p^{antidote}$ promoter is defective at killing spores that do not inherit the allele encoding the Wtf4$^{poison}$. This is surprising given the high sensitivity of mitotic *S. pombe* cells to even low levels of Wtf4$^{poison}$ we observed in previous work [52] and in this study (e.g., S9 Fig Diploid 24). We hypothesized that the Wtf4$^{antidote}$ is better at preventing Wtf4$^{poison}$ from entering spores when the proteins were co-expressed from the p$^{antidote}$ promoter than when the Wtf4$^{poison}$ is expressed from its endogenous promoter. To test this possibility, we compared the amount of Wtf4$^{poison}$ signal in spores produced by diploids heterozygous for both $p^{antidote}$-wtf4$^{poison}$-GFP allele (at *ade6*) and wtf4$^{antidote}$-mCherry (at *ura4*) to similar control diploids in which wtf4$^{poison}$-GFP was expressed

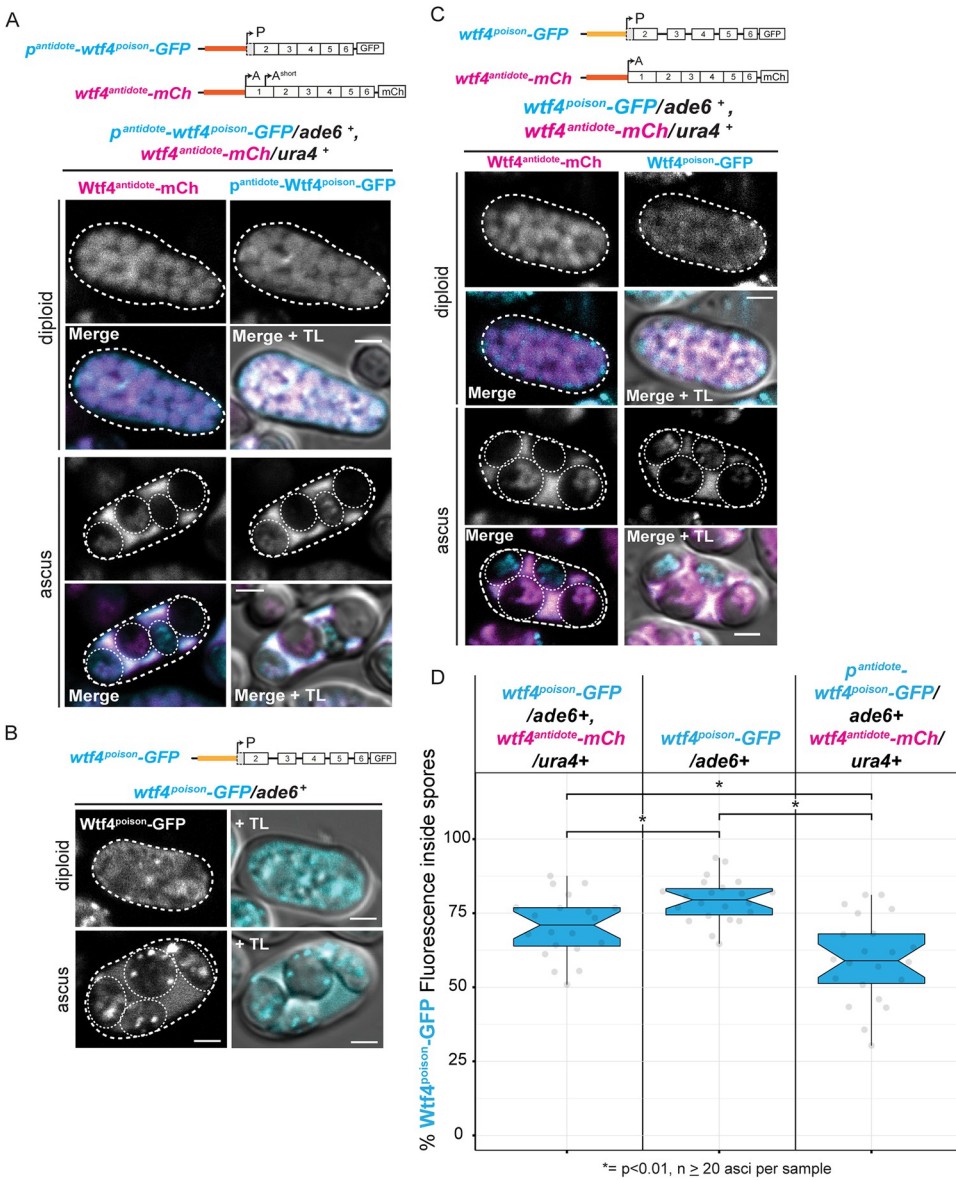

**Fig 5. Wtf4$^{poison}$ localization is altered when expressed from the $p^{antidote}$ promoter.** (**A**) Image of a $p^{antidote}$-$wtf4^{poison}$-GFP/ ade6+, $wtf4^{antidote}$-mCherry/ura4+ diploid and tetratype ascus. $p^{antidote}$-Wtf4$^{poison}$-GFP is shown in cyan and Wtf4$^{antidote}$-mCherry is shown in magenta in merged images. Additional images (parental ditype and non-parental ditype) can be found in S9B Fig. (**B**) Image of a $wtf4^{poison}$-GFP/ ade6+ diploid and ascus. Images of this strain were first presented in [37]. (**C**) Images of a heterozygous $wtf4^{antidote}$-mCherry/ura4+, $wtf4^{poison}$-GFP/ade6+ diploid and ascus. Wtf4$^{antidote}$-mCherry is shown in magenta and Wtf4$^{poison}$-GFP is shown in cyan in merged images. Images of this cross are also depicted in Fig 1E. (**D**) Quantification of the percent of total ascal Wtf4$^{poison}$-GFP fluorescence located outside spores in asci generated by the following diploids: 1. $wtf4^{poison}$-GFP/ade6+, $wtf4^{antidote}$-mCh/ura4+, 2. $wtf4^{poison}$-GFP/ade6+, and 3. $p^{antidote}$-$wtf4^{poison}$-GFP/ade6+, $wtf4^{antidote}$-mCh/ura4+ (* = p < 0.01, t-test, n> 20 asci per sample). TL = transmitted light. All scale bars represent 2 μm. All images were acquired after 2 days on sporulation media. Images were taken using the same settings. Not all images are shown at the same brightness and contrast to avoid over saturation of pixels in the brighter images.

from its endogenous promoter (Fig 5A and 5B). Consistent with our hypothesis, we found that less of the total Wtf4$^{poison}$-GFP signal was inside of spores when the protein was expressed from the $p^{antidote}$ promoter, as compared to when the protein was expressed from the endogenous $p^{poison}$ promoter (59 vs 71%). We also noticed that less Wtf4$^{poison}$-GFP (from the

endogenous promoter) was included spores in the presence of the Wtf4$^{antidote}$ protein (71 vs 79%), further supporting that the Wtf4$^{antidote}$ plays a role in excluding Wtf4$^{poison}$ from spores (Fig 5B–5D).

## Discussion

### Dual promoters with distinct regulation are key to *wtf* meiotic drive

In *wtf4* drive, all spores are exposed to the Wtf4$^{poison}$, while only those that inherit the *wtf4* allele generally receive a sufficient dose of the Wtf4$^{antidote}$ to survive. Previously, we observed Wtf4$^{poison}$ in meiotic cells prior to spore formation, but observed Wtf4$^{antidote}$ enriched only in spores that inherited *wtf4* [37]. These observations explained how all spores were poisoned and how only those that inherited the *wtf4* locus were rescued by the antidote, but it raised a new puzzle of how meiotic cells survived exposure to the Wtf4$^{poison}$ in the apparent absence of the Wtf4$^{antidote}$. This uncertainty was amplified by subsequent work that showed Wtf4$^{poison}$ efficiently kills cells, not just spores, that do not express antidote [52]. In this work, our new data offer a solution to the puzzle in that we found there is also Wtf4$^{antidote}$ expression in cells undergoing meiosis. This result is consistent with previous long-read RNA sequencing data that show a low level of transcription of *wtf*$^{antidote}$ alleles in early meiosis [59].

Our work also reveals that different transcriptional regulation of the poison and antidote promoters is largely responsible for the distribution of the Wtf4$^{poison}$ to all spores and the enrichment of the Wtf4$^{antidote}$ in the spores that inherit the locus, as transcriptional reporters grossly recapitulate localization patterns of tagged proteins (Fig 2). However, different features of the proteins also contribute to their localization patterns. Specifically, we found that the propensity for the two Wtf proteins to be included in developing spores was different. The Wtf4$^{antidote}$ present in the cytoplasm at the time of spore encapsulation is excluded from the spores more than Wtf4$^{poison}$ present at the same time (Fig 4D). In addition, our results suggest that Wtf4$^{antidote}$ protein can promote exclusion of the Wtf4$^{poison}$ from spores as they individualize, as a smaller fraction of the Wtf4$^{poison}$ enters spores in the presence of the Wtf4$^{antidote}$ (Figs 4D and 5D). This change in localization of the Wtf4$^{poison}$ in the presence of the Wtf4$^{antidote}$ is consistent with our previous observations that the Wtf4 proteins co-assemble and are trafficked to the vacuole [52].

Our data suggest the following model (Fig 6): the Wtf4$^{poison}$ is mostly expressed prior to spore formation and then subsequently packaged in spores (e.g., Fig 1E). This likely ensures that each spore gets a lethal dose from a heterozygous zygote. Conversely, Wtf4$^{antidote}$ is present prior to spore formation, perhaps to prevent the cells undergoing meiosis from succumbing to the poison (e.g., Fig 1E). The antidote that is present at the time of spore formation likely offers little protection to spores that do not inherit the driving locus because it does not prevent all Wtf4$^{poison}$ from entering spores (Fig 5D) and the Wtf4$^{antidote}$ is excluded from developing spores (Fig 4). We speculate that the exclusion of Wtf4$^{antidote}$ from spores is due to its association with vacuoles, which are also excluded from developing spores [52,67]. Because of these factors, spores that inherit *wtf4* must produce their own private supply of Wtf4$^{antidote}$, while those that do not succumb to the poison.

### Difficulties in universal suppression of *wtf* drivers

This work also adds to our understanding of the evolutionary success of the *wtf* drivers within *Schizosaccharomyces* species [64]. In *S. pombe*, all sequenced isolates contain between 25 and 38 *wtf* genes, including 4–14 genes that are predicted or shown to be intact meiotic drivers [35–38]. Each heterozygous driver can cause the destruction of half the spores and antidote proteins generally do not protect against poisons that have distinct sequences [35,38]. Because of these features, diploid zygotes produced by crossing two diverged isolates produce very few

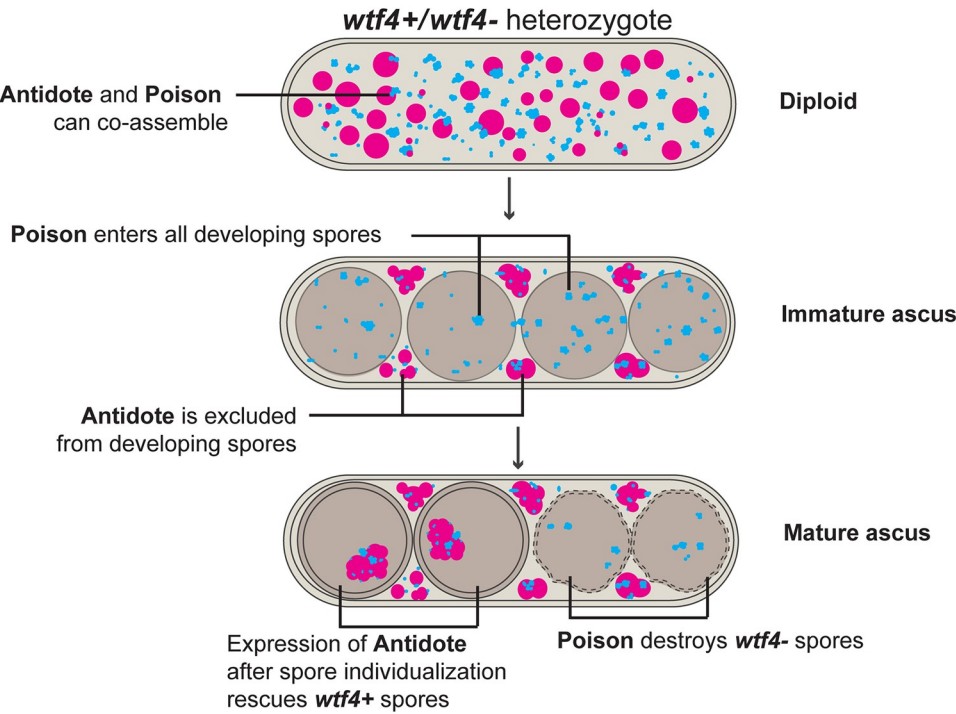

**Fig 6. Model of *wtf4* drive in *wtf4+/wtf4-* heterozygote.** The Wtf4 proteins are both present prior to spore formation (top). As spores form (middle), more Wtf4antidote is excluded from spores relative to Wtf4poison. Additional Wtf4antidote is produced in *wtf4+* spores (bottom), which rescues them from the Wtf4poison. The *wtf4-* spores are destroyed by the Wtf4poison.

viable spores [35,38,43–45,68,69]. The incentive to evolve a suppression mechanism to suppress a *wtf* driver, therefore seems quite high [70].

There are likely many roads to suppression of *wtf* drivers. We speculate, however, that none of them have been terribly successful given the *wtf* gene family has been causing meiotic drive for over 100 million years [64]. The known genic suppressors of *wtf* drivers are all Wtfantidote proteins. Importantly, Wtfantidote proteins seem highly specific in that they neutralize only Wtfpoison proteins with very similar sequences [35,38,39,52]. Because of this, Wtfantidote proteins are not expected to act as general suppressors of *wtf* drivers, as the genome would require a unique suppressor for each driver. A non-Wtf universal suppressor that interfaces with Wtfpoison proteins might also be difficult to envision given the vast diversity of Wtf proteins, which can share as little as 30% amino acid identity within a species [35,36].

Transcriptional repression seems like a more feasible route to universal suppression of *wtf* drivers given the conservation of their promoters ([36,38,60]; Fig 2A and 2D). Previous work has identified some factors that help regulate the expression of *wtf* genes, however, these studies did not distinguish between *wtf*poison and *wtf*antidote transcripts [46,48]. In mitotic cells, a mutation in the histone deacetylase *clr6* or inhibition of histone deacetylation with the drug Trichostatin A leads to increased *wtf* gene transcription [46–48,71]. Also in mitotic cells, *wtf* mRNAs can be eliminated from cells via double-stranded-directed RNA decay [72]. It is possible that these mitotic mechanisms may affect *wtf* gene expression during meiosis as well, but this has not been determined. In meiotic cells, Cuf2 decreases the expression of *wtf* genes late in meiosis [51,73]. However, this transcriptional suppression appears to have a minimal effect on drive. For example, Cuf2 decreases expression of *wtf13*, but even in this suppressed state, *wtf13* drives into >90% of the gametes produced by a *wtf13+/wtf13Δ* heterozygote [39].

It is important to note that not all *wtf* genes are equally expressed, and differential expression of the *wtf*<sup>*poison*</sup> transcripts is correlated to drive strength in *S. octosporus* [37,38,59,64]. This differential expression could reflect an inefficient transcriptional repression system, but that has not been investigated. Still, effective transcriptional repression of all *wtf* drivers could be challenging for several reasons. The first challenge is the large number of *wtf* genes and the fact that they are not all found at the same locus. This may make it hard to simply turn them all off *en masse* by packaging one genomic region in heterochromatin. Even if the genes were found in one location, the genomic region housing the *wtf* genes would likely be under strong selection to resist such heterochromatization. Specifically, haplotypes that could avoid heterochromatization and thus better maintain drive would be selected by drive. This is because regions linked to meiotic drivers also get to enjoy the benefits of drive and thus profit from enhancing drive, rather than suppressing it [70]. This selective pressure resisting heterochromatization would also apply to the distributed *wtf* drivers as well. This situation is profoundly different than loci housing transposable elements as those selfish elements generally offer no evolutionary advantages to flanking sequences [2].

Another challenge in establishing suppression of *wtf* drivers is that partial or imperfect suppression of *wtf* genes may do more harm than good. For example, if silencing is incomplete in a diploid homozygous for a given driver, only one allele could be expressed, and drive would occur in a cell where no drive would occur in the absence of silencing. In addition, in some instances *wtf* drivers are suppressed by other *wtf* genes [35,39]. This is an additional situation in which incomplete silencing of the wrong *wtf* gene could lead to more, rather than less, drive.

An ideal suppression approach would be to suppress a key transcription factor driving expression from *p*<sup>*poison*</sup> promoters. This approach would allow the cell to stop drive, but it also comes with minimal risk as partial suppression would still be selectively advantageous by decreasing the likelihood of drive. In this scenario, an imperfect system could adapt and improve over time. In this work, we identified Mei4 as the key transcription factor governing the expression of *p*<sup>*poison*</sup> (Fig 3). Suppression of Mei4, however, is not an ideal option for suppressing drive because it is an essential regulator of meiosis. Mei4 controls the expression of over 100 genes and cells lacking Mei4 fail to complete meiosis [50,51,62,63,74–77].

## Final speculation

It is hard not to admire how beautifully adapted the *wtf* parasites are at promoting themselves despite the costs. Equally remarkable is how well insulated *wtf* parasites appear to be against their host genomes acting to stop their expression. That is not to say, however, that *Schizosaccharomycetes* have not found other ways to mitigate the costs of *wtf* drivers. Some *S. pombe* natural isolates preferentially inbreed and some isolates generate disomic (aneuploid or diploid) spores at high frequencies (up to 46% of spores; [41,78]). Although it is hard to say if the *wtf* genes promoted the evolution of these traits, it is possible, as both traits effectively suppress fitness costs of *wtf* drivers [41,78]. If these traits were selected for based on their ability to suppress drive, suppression likely came at a steep price to fitness, given that inbreeding and meiotic disruption are generally not good for fitness. Perhaps this was the price that had to be paid, as cheaper options, like universal transcriptional silencing, were out of reach.

## Materials and methods

### Generation of yeast strains

All yeast strains, plasmids, and oligos are described in S1–S3 Tables. We confirmed the plasmids generated in this section via Sanger sequencing. We completed all transformations

described in this section using standard lithium acetate protocol [65], first selecting for drug resistance and then screening for the relevant auxotrophy.

**Generation of an allele of *wtf4* with endogenous promoters and *mCherry-wtf4^poison^*.**
We ordered a gBlock from Integrated DNA technologies (IDT, Coralville IA) containing the $p^{antidote}$ promoter (-600 bp), exon 1, intron 1, and sequence encoding an mCherry tag and a 5X glycine linker upstream of the $wtf4^{poison}$ start codon [79]. The gBlock also included 177 bp of exon 2. We amplified this gBlock using oligos 620 and 718. We also amplified the rest of the *wtf4* sequence from pSZB189 using oligos 679 and 687 [37]. We then used overlap PCR to stitch the PCR fragments together using oligos 620 and 687. Next, we digested the resulting PCR product with SacI and ligated it into the SacI site of pSZB188 [37], an *ade6* integrating vector with a *kanMX4* cassette, to create pSZB250. We cut pSZB250 with KpnI and transformed the linearized vector into SZY643 to generate SZY1037.

**Generation of a *wtf4* allele that encodes only mCherry-Wtf4^poison^ under the control of the endogenous promoter.**   This allele has the full-length *wtf4* gene (including $p^{antidote}$ promoter) but the two start codons in exon 1 (for the Wtf4^antidote^) are mutated to TAG. We first cloned pSZB259 which contains an untagged *wtf4* allele in which both start codons are mutated, similar to pSZB257 [37]. We amplified the $p^{antidote}$ promoter and the mutated version of exon 1 from pSZB259 using oligos 620 and 861. We amplified the $mCherry-wtf4^{poison}$ sequence from pSZB250 (described above) using oligos 862 and 687. We then used overlap PCR to stitch the two PCR products together using oligos 620 and 687. We digested the resulting PCR product with SacI and ligated the cassette into the SacI site of pSZB188 [37] to make pSZB355. We also digested pSZB355 with SacI to isolate the *wtf4 allele* and cloned it into SacI-digested pSZB386, a hygromycin B-resistant *ade6* integrating vector [37] to generate pSZB824. We digested pSZB824 with KpnI to linearize the construct and transformed into SZY2572 to generate SZY4570.

**Generation of a *wtf4* allele that encodes only Wtf4^antidote^-GFP under the control of the endogenous promoter.**   We used pSZB203 as a template to amplify the 5' end of *wtf4-GFP* with oligos 620 and 736 and the rest of the gene with oligos 735 and 634 [37]. The two PCR products were then stitched together using overlap PCR using oligos 620 and 634. The 735 and 736 oligos mutated the start codon for the $wtf4^{poison}$ to TAC. The resulting PCR product was then digested with SacI and ligated into the SacI site of pSZB188 to generate pSZB260. We cut pSZB260 with KpnI and integrated it into the *ade6* locus of SZY643 to create SZY1056.

**Generation of a *wtf4* allele that encodes only mCherry-Wtf4^antidote^ under the control of the endogenous promoter.**   We used site-directed mutagenesis to mutate the translational start site from ATG to TAC of $wtf4^{poison}$ within pSZB248 [37] to generate pSZB367. We linearized pSZB367 with KpnI and integrated it into SZY643 to make SZY3586.

**Generation of a *wtf4* allele with the first antidote start site mutated to make the $wtf4^{short}$ allele.**   This allele has the full length *wtf4* gene, with the first antidote start site (ATG) mutated to TAG. Using pSZB189 as a template, we amplified the 5' end of *wtf4* with the mutated start site using oligos 620 and 702. We then amplified the rest of the *wtf4* gene using oligos 701 and 686 with pSZB189 as the template. The two PCR products were then stitched together using overlap PCR using oligos 620 and 686 and ligated into the SacI site of pSZB188 to generate pSZB244. We cut pSZB244 with KpnI and integrated it into the *ade6* locus of SZY643 to create SZY1026.

**Generation of a p^antidote^-mCherry transcriptional reporter.**   We first amplified the $wtf4^{antidote}$ promoter linked to mCherry using pSZB248 [37] as a template with oligos 688 and 1447. Next, we amplified the *ADH1* transcriptional terminator from pKT127 [80] using oligos 1448 and 634. We then used overlap PCR to unite these fragments and generate $p^{antidote}$-mCherry. We then digested the complete $p^{antidote}$-mCherry cassette with SacI and ligated it into

the SacI site of pSZB386 [39] to create pSZB766. We cut pSZB766 with KpnI and integrated it into the *ade6* locus of SZY2080 to create SZY2137. We also digested the cassette form pSZB766 and cloned it into SacI-digested pSZB331 [35] to create pSZB744. We cut pSZB744 with KpnI and integrated it into the *ura4* locus of SZY2080 to create SZY4534.

**Generation of a p$^{antidote\ long}$-mCherry transcriptional reporter.** We amplified the majority of the *p$^{antidote\ long}$-mCherry* construct (with an *ADH1* transcriptional terminator) using oligos 3024 and 634 and pSZB891 [52] as a template. We then added the rest of the upstream to the *p$^{antidote\ long}$-mCherry* construct by using it as a template for PCR with oligos 3025 and 634. This generated the full *p$^{antidote\ long}$-mCherry* construct. We digested this cassette with SacI and ligated it into the SacI site of pSZB322 [39] to create pSZB1361. We cut pSZB1361 with KpnI and integrated into the *lys4* locus of SZY2080 [52] to generate SZY4442.

**Generation of a strain with *wtf4::hphMX6* at the *ade6* locus.** We digested pAG32 (*Goldstein and McCuster, 1999*) with NotI to isolate the *hphMX6* cassette. We transformed the *hphMX6* cassette into SZY887 [37] selecting for HYG resistance and lack of G418 resistance. This created SZY969.

**Generation of a p$^{poison}$-GFP transcriptional reporter.** We used pSZB203 as a template to amplify the *p$^{poison}$* promoter using oligos 1174 and 1549, and to amplify GFP (with an *ADH1* transcriptional terminator) using oligos 1548 and 634 [37]. We used overlap PCR to join these two pieces using oligos 1174 and 634. We then digested the complete *p$^{poison}$-GFP* construct with SacI and cloned it into SacI-digested pSZB386 to generate pSZB821. We linearized pSZB821 with KpnI and integrated it into SZY643 to make SZY2279.

**Generation of an allele of *wtf4$^{antidote}$-GFP* in which the p$^{antidote}$ promoter is replaced with the p$^{poison}$ promoter.** We amplified the poison promoter and exon 1 from pSZB553 (see above) using oligos 1174 and 604. We amplified the rest of the *wtf4* coding sequence from pSZB700 [52] using oligos 605 and 997. We amplified GFP (with an *ADH1* transcriptional terminator) from pSZB203 [37] using oligos 998 and 634. We used overlap PCR to join the three pieces using oligos 1174 and 634. We digested the complete *p$^{poison}$-wtf4$^{antidote}$-GFP* construct with SacI and cloned it into SacI-digested pSZB386 to generate pSZB727. We digested pSZB727 with KpnI to linearize the construct and transformed into SZY44 to generate SZY2406.

**Generation of the *p$^{antidote}$-wtf4$^{poison}$-GFP* allele.** We amplified *p$^{antidote}$* from pSZB203 using oligos 688 and 1383 [37]. We amplified the *wtf4$^{poison}$* coding sequence from pSZB392 [52] using oligos 1384 and 997. We amplified GFP (with an *ADH1* transcriptional terminator) from pSZB203 [37] using oligos 998 and 634. We used overlap PCR to join the three pieces using oligos 688 and 634. We digested the complete *p$^{antidote}$-wtf4$^{poison}$-GFP* construct with SacI and cloned it into SacI-digested pSZB386 to generate pSZB758. We digested pSZB758 with KpnI to linearize the construct and transformed into SZY2572 to generate SZY4525. The *p$^{antidote}$-wtf4$^{poison}$-GFP* allele lacks exon 1 and thus does not encode Wtf4$^{antidote}$.

**Generation the *wtf4$^{FLEXΔ}$-GFP* allele.** We used site-directed mutagenesis to delete the FLEX motif (TTTGTTTAC, [62,63]) within intron 1 of the *wtf4-GFP* allele in pSZB203 [37]. This generated pSZB848. We digested pSZB848 with KpnI to linearize the construct and transformed into SZY643 to generate SZY1479.

**Generation of an *ade6+::his5Δ* allele.** This was completed in the same manner as in [37]. Briefly, we amplified from genomic DNA a region upstream of *his5* using oligos 795 and 796, a region downstream of *his5* using oligos 797 and 798, and *ade6+* using oligos 799 and 800. We stitched these pieces together using overlap PCR and oligos 795 and 798. We then transformed the cassette into SZY631 [39], selecting for *ade6+* and then screening for *his5-*. This generated SZY1285 used in this work.

**Generation of a *ura4+, wtf4^poison^-GFP* strain.** We amplified a *ura4+* cassette using oligos 34 and 37 and SZY44 [37] as a template. We then transformed it into SZY1049 [37], selecting for *ura4+*, to generate SZY5047.

**Generation of a *pat-1as(L95G), wtf4^poison^-GFP* strain.** We crossed SZY44 with J1687 [63] and selected for progeny that were resistant to hygromycin and G418, but were sensitive to nourseothricin, to obtain SZY5581. We then transformed KpnI-digested pSZB257 [37] into SZY5581, which resulted in SZY5638, a strain with *pat1.L95G and wtf4^poison^-GFP* alleles.

**Generation of a *pat1.L95G, wtf4^antidote^-mCherry* strain.** We transformed KpnI-digested pSZB891 [37] into SZY5581 to obtain SZY5685, a strain with the *pat1.L95G and wtf4^antidote^-mCherry* alleles.

**Generation of a diploid with *pat1.L95G, wtf4^antidote^-mCherry, wtf4^poison^-GFP*.** We generated *h-/h-* diploids (SZY5742 and SZY5743) by protoplast fusion of SZY5685 and SZY5638 as previously described [81], with the following changes: We grew 50 mL YEL cultures of each strain to mid-logarithmic phase (~5x10^6 cells/ml) before washing in 10 mL 0.65 M KCl. We then prepared protoplasts by incubating cell pellets in 0.65 M KCl containing 0.1 g/mL Lall-zyme MMX (Scott Labs) for 14 minutes. We then plated the protoplasts to YNP -adenine, -his-tidine, -leucine, -uracil and incubated at 32°C for 3 days. We then replica plated the colonies and imaged the cells to confirm genotypes and ploidy.

## Allele transmission and viable spore yield

All allele transmission and viable spore yield (VSY) assays were completed as previously described [37,53]. Briefly, we generated stable diploids by mixing each haploid parent in a microcentrifuge tube and plating them on SPA (1% glucose, 7.3 mM KH$_2$PO$_4$, vitamins, agar) for ~15 h at room temperature to allow the cells to mate. We scraped the mated cells off of SPA and spread on a medium to select for heterozygous diploids (minimal yeast nitrogen base plates). We grew diploid colonies overnight in 5 mL of rich YEL broth (0.5% yeast extract, 3% glucose, 250 mg/L of ade-nine, lysine, histidine, and uracil). We then plated ~100 mL onto SPA to induce sporulation, as well as diluted samples onto YEA (same as YEL, but with agar). We confirmed the colonies that grew on the YEA plate were truly heterozygous diploid cells by replicating to diagnostic media and counted the colonies for VSY. After 3 days, we scraped the cells from the SPA plates, treated it with glusulase (Sigma (G7017-10ML) and ethanol to isolate spores, and plated dilutions of the spores on YEA [53]. We then counted and phenotyped the spore colonies using standard approaches. At least 2 diploids were assayed per cross and at least 150 spores were genotyped for allele transmission assays. The raw data can be found in the S1 Data file.

## Alignment of promoters

For the alignment of antidote promoters (Fig 2A), we aligned the 800 base-pairs upstream of 41 predicted antidote-only alleles [36] from three different strains of *S. pombe* (the reference genome, *S. kambucha*, and *FY29033, Lock et al., 2018*). For the alignment of poison promoters (Fig 2D), we aligned the intron 1 sequences (flanked by sequences 100bp upstream and down-stream intron 1) of 28 predicted poison-antidote *wtf* drivers [36] from three different strains of *S. pombe* (the reference genome, *S. kambucha*, and *FY29033*). We utilized Geneious (version 11.0.14.1, https://www.geneious.com) using the Geneious aligner with the "global alignment without free end gaps" setting, taking the percent identity at each nucleotide position.

## Protein analysis with western blots

To isolate proteins for *pat1+* western blots (S4 and S5 Figs), we grew diploids ~20 hours to sat-uration in 10 ml YEL at 32°C. We then diluted cells 1:200 in 100 mL PM media and again

grew at 32°C overnight (20–24 hours). We then washed the cells once in PM-N and then diluted to an O.D. 600 of 1.0 (~1x10$^7$ cells/ml) in 500 ml PM-N incubated at 25°C. We prepared whole cell extracts at the time-points indicated in figure legends from 100 mL aliquots as described in [82] with the following addition. To block protein degradation, we added 1 mM PMSF to each time-point prior to spinning down cells. The cell lysis buffer (50 mM Tris-HCl pH 7.5, 150 mM NaCl, 1 mM EDTA, 0.5% NP-40, 10% Glycerol) we used was supplemented with 1 mM PMSF and 1 complete Mini, EDTA-free protease inhibitor cocktail tablet per 5 mL. We disrupted the cells using a Mini Bead beater (Biospec Products) and we imaged cells to ensure spores were broken.

To prepare positive controls expressing constitutive GFP or mCherry, we grew strains SZY2636 (GFP) and SZY2638 (mCherry) [78] to saturation in YEL, then pelleted a 1 mL volume, washed the cells in ddH$_2$0, resuspended in 1 mL ddH$_2$0 and plated 200 μl onto each of three SPA+S plates. We then incubated the plates at 25°C for three days and then scraped cells into 1 mL ice-cold PBS and prepped whole cell extracts as described above. We imaged the cells to ensure sporulation using a Zeiss Observer.Z1 wide-field microscope with a 63x 1.4 Oil DICII objective and collected the emission onto a Hamamatsu ORCA Flash 4.0 using μManager software. We used BP 440–470 nm to excite GFP and collected BP 525–550 emission using a FT 495 dichroic, and mCherry with BP 530–585 nm excitation and LP 615 emission, using an FT 600 dichroic filter.

To run western blots, we diluted samples 1:1 in LDS sample buffer (Life Technologies) and heated them for 10 min at 75°C before loading. We then ran the proteins on NuPAGE 4–12% Bis-Tris protein gels (Life Technologies) in 1 x MOPS buffer (Life Technologies) for 50 minutes at 200V and then transferred to PVDF membranes (Bio-RAD #1704156) using the Trans-Blot Turbo system. We probed the membranes with either a rabbit monoclonal, α-GFP antibody (Cell Signaling Technology #2956) at 1:1000 and/or a mouse monoclonal, α-mCherry antibody (Millipore, MAB131873) at 1:1000, overnight at 4°C with agitation in Odyssey blocking buffer (TBS, from LI-COR biosciences). Secondary, α-rabbit (800 cW) and/or α-mouse antibodies (680 cW) were used for fluorescent visualization of the proteins. We imaged the blots on the Odyssey-CLx (LI-COR biosciences). To show protein loading we either stained membranes with Ponceau S (Cell Signaling Technology #59803) or ran a second gel with the same volume of sample and stained with Imperial protein stain (Thermo Scientific #24615).

## Synchronization of *pat1.L95G* cultures

To generate synchronous meiotic cultures (S6 Fig), we grew *h-/h- pat1.L95G/pat1.L95G* analog-sensitive diploid strains (SZY5742 and SZY5743) for 20 hours at 30°C in 10 mL YEL media. We then diluted the cells 1:250 in 50 mL EMM supplemented with 75 μg/ml lysine and grew them for 20–24 hours at 30°C. We then washed the cells twice in EMM-N and diluted to an O.D. 600 of 1.0 in EMM-N supplemented with 10 μg/ml lysine. After 12 hours at 30°C, we added 50 μg/ml lysine, 500 μg/ml NH4Cl and 25 μM 3MB-PP1 (Cayman Chemicals) to each culture, and then sampled cells for DAPI staining, imaging and Westerns at the timepoints described in the figures.

## DAPI staining

We pelleted a 1 ml volume of each culture at the indicated timepoints and resuspended the cells in 1 ml 70% ethanol. We then incubated the cells at room temperature for one hour and then washed once in 1 ml PBS. We then immediately visualized cells using a Zeiss Observer.Z1 wide-field microscope with a 63x 1.4 Oil DICII objective and collected the emission onto a

Hamamatsu ORCA Flash 4.0 using μManager software. We excited DAPI at 365 nm and fluorescence was collected through a 445/50 nm bandpass filter.

## Fluorescence microscopy

The raw data for all image quantification is presented in the S2 Data. We generated diploids as previously described [37] and placed them on sporulation agar (SPA, 1% glucose, 7.3 mM $KH_2PO_4$, vitamins, agar) for 1–3 days. We then scraped the cells off the plates and onto slides for imaging. For the *pat1.L95G* time course experiment shown in S6 Fig, we fixed the cells in 4% paraformaldehyde for one hour at room temperature, washed three times in 1 mL PBS and stored the cells at 4˚C prior to imaging. For all microscopy, except for the experiments listed below, we used an LSM-780 (Zeiss) microscope, with a 40x C-Apochromat water-immersion objective (NA 1.2), in photon-counting channel mode with 488 and 561 nm excitation. We collected GFP fluorescence through a 481–552 bandpass filter and mCherry through a 572 long-pass filter. We also used photon-counting lambda mode, with 488 and 561 nm excitation, collecting fluorescence emission over the entire visible range. We then used these images to linearly unmix the fluorescence spectra using an in-house custom written plugin for ImageJ (https://imagej.nih.gov/ij/) to verify that there was no auto-fluorescence in the cells. Brightness and contrast are not the same for all images. We utilized two independent progenitor diploids and assayed at least 25 asci for each genotype represented. We called an ascus "mature" if all four spores showed distinct, dark outlines via transmitted light, suggesting spore membranes had been formed. For images where fluorescence intensity inside of spores is compared to fluorescence intensity outside of spores, great care was taken in acquiring the data so that only asci where all spores were sharply in focus were used for comparison.

For the *wtf4^antidote^-mCherryh/ura4+* (S2C Fig) we imaged using a Zeiss Observer.Z1 wide-field microscope with a 40x C-Apochromat (1.2 NA) water-immersion objective and collected the emission onto a Hamamatsu ORCA Flash 4.0 using μManager software. We used BP 440–470 nm to excite GFP and collected BP 525–550 emission using a FT 495 dichroic, and mCherry with BP 530–585 nm excitation and LP 615 emission, using an FT 600 dichroic filter, and mCherry with BP 530–585 nm excitation and LP 615 emission, using an FT 600 dichroic filter. Brightness and contrast are not the same for all images.

For the gametogenesis time-lapse imaging (Fig 1C–1F), we crossed a haploid *Sp* strain carrying *mCherry-wtf4* (SZY1142) to one with *wtf4^poison^-GFP* (SZY5047) to generate heterozygous diploids as previously reported [37]. We also crossed a haploid *Sp* strain carrying *wtf4^antidote^-mCherry* (SZY2572) to one with *wtf4^poison^-GFP* (SZY5047) to generate heterozygous diploids using the same method as above. We grew these diploids to saturation in 5 mLs of rich YEL broth (0.5% yeast extract, 3% glucose, 250 mg/L of adenine, lysine, histidine, leucine, and uracil) overnight at 32˚C. We then diluted 100 μL of these diploid cultures into 5 mLs of PM media (20 mLs of 50x EMM salts, 20 ml 0.4 M $Na_2HPO_4$, 25 mL 20% $NH_4Cl$, 1 mL 1000x vitamins, 100 μL 10,000x mineral stock solution, 3 g potassium hydrogen phthalate, 950 mL $ddH_2O$, 25 mL of sterile 40% glucose after autoclaving, supplemented with 250 mg/L uracil) and again grew overnight at 32˚C. The next day, we spun to pellet and resuspended the pellet in PM-N media (PM without $NH_4Cl$). We shook the PM-N cultures for 4 h at 28˚C. We loaded the plate with PM-N media after washing and flushing the microfluidic plate with PBS. Diploids were trapped in an EMD Millipore CellASIC (Haploid budding yeast) microfluidic plate. PM-N media was flowed through the plate for the duration of the movie. The temperature was maintained through the CellASIC manifold at 25˚C. GFP and mCherry fluorescence were excited with a Spectra III illuminator with a 475 nm and 555 nm dichroic, respectively. The fluorescence emission was then collected with a 515/30 nm emission filter for GFP and a 595/

31 nm emission filter for mCherry. The fluorescence was recorded by a Prime 95B sCMOS camera (Photometrics). Images from multiple positions were recorded every 10 minutes for 24 hours. Each experiment had two biological replicates of each diploid and was imaged at 4 XY locations analyzed in parallel. We repeated the experiments with both diploids twice. The total number of diploid cells analyzed were n = 25 for *mCherry-wtf4/wtf4$^{poison}$-GFP* diploids and n = 13 for *wtf4$^{antidote}$-mcherry/wtf4$^{poison}$-GFP* diploids.

Analysis of these data was performed in Fiji (https://imagej.net/software/fiji/) using a few different plugins written in house (https://research.stowers.org/imagejplugins/). First stage drift was eliminated with "stackregj". The data was then smoothed with a 1-pixel Gaussian blur. Next, the background was subtracted with a 50-pixel rolling ball. After this, ROIs were drawn by hand around each cell of interest. The average intensity over time was plotted using "create spectrum jru v1". Time traces for GFP and mCherry for each cell were recorded separately. Next, each time trace was aligned in time manually by marking the first frame (from transmitted light only) that spores started to appear and setting that frame to time zero with "set multi plot offsets jru v1". Then, all time traces for a particular strain and for GFP and mCherry only were combined into one plot with "combine trajectories jru v1". These plots were then normalized to the minimum and maximum using "normalize trajectories jru v1". The traces were then imported to Prism 9 (https://www.graphpad.com/scientific-software/prism/) using the XY graph preset, with the mean and 95% confidence interval plotted against time. The time scale for the plots was recalculated using the frame rate of the time lapse.

For the Fluorescence Recovery After Photobleaching experiments, cells were imaged on a Ti2-E (Nikon) microscope coupled to a CSU-W1 Spinning Disc (Yokogawa). GFP and mCherry were laser excited at 488 nm and 561 nm, respectively, through a 60x (spore FRAP NA 1.4) or 100x (whole ascus FRAP NA 1.45) Plan Apochromat objective. The fluorescence emission of GFP was collected through a 525/36 nm filter, while the emission of mCherry was collected through a 605/25 nm filter. Both signals were collected on a Prime 95B camera (Photometrics).

For the spore and whole ascus FRAP experiments, an initial frame of fluorescence was recorded in each channel. Then, many spores or asci were bleached to background in the same large field of view, using 100% power from the 450nm, 550nm and 640nm laser lines. Once bleaching was completed, the recovery was recorded every 10 minutes for ~6 hours total time (whole FRAP) or every 5 minutes for 4 hours total time (spore FRAP). To generate the recovery curves, using Fiji, the average fluorescence intensity was quantified inside all spores containing GFP signal and all spores containing mCherry signal per frame, separately. Once the curves were obtained, all curves of the same fluorophore were normalized to the min and max intensity values and averaged to yield the final curves.

To quantify the amount of fluorescence intensity inside and outside spores, asci were imaged on an LSM 780 (Zeiss) in photon counting mode with a 40x LD C-Apochromat Objective (NA 1.1). GFP and mCherry were excited at 488 and 561nm, respectively. GFP fluorescence was collected through a 482–553 nm bandpass filter and mCherry fluorescence was collected through a 572–735 nm bandpass filter. Single z slices were collected with transmitted light. Then for each ascus, an ROI was drawn around the entire ascus, and around each of the 4 spores individually (drawn on the transmitted light without input from the fluorescence). This occurred either manually or via Cellpose (https://www.cellpose.org/) with manual editing of the auto generated ROIs. Then the sum-total intensity of the ascus in each channel was measured. Then the sum total intensity of each spore in each channel was also measured. For each channel, the 4 spores' intensities were added together and divided by the total intensity to yield the fractional spore intensity. This was done for each channel individually. The results of many fractional spore intensities were plotted together.

### Analysis of published Mei4 Chip-Seq data

FASTQ data from [50] was retrieved from the European Nucleotide Archive (ENA), with accession number: ERP001894. Data was trimmed for quality using Trimmomatic [83] and was aligned to *S. pombe* genome version ASM294v2 with bowtie2 [84] using default parameters. Peaks were called using MACS2 with callpeak -gsize $1.38^{e7}$ and -q 0.05. Metagene plots were constructed by averaging the RPM (Reads Per Million) signal for individual genes (either the predicted *wtf* meiotic drive genes (*wtf4*, *wtf13*, top*)* or the *wtf* genes that are predicted to only encode antidote proteins (*wtf5*, *wtf9*, *wtf10*, *wtf16*, *wtf18*, *wtf20*, *wtf21*, *wtf25*, bottom)) across 400 bins and then averaging multiple genes to create a single profile followed by loess smoothing with a span of 0.05. For any reads that mapped to more than one location, only a single location, chosen at random, is reported.

## Supporting information

**S1 Fig. Representation of alleles used in this study.** Each allele is represented as a cartoon with the allele numbers referenced through the text. The depictions describe the promoters (orange for $p^{antidote}$ and yellow for $p^{poison}$), the translational start site of the proteins (black arrows), fluorescent tags (mCherry or GFP), translational start site mutations (red stars) and each protein produced. Wtf4$^{antidote}$ is represented as "A" and Wtf4$^{poison}$ is represented as "P". We also depict the FLEX motif (black box) found within $p^{poison}$. These depictions are also present in figures where each of these alleles is used.
(TIF)

**S2 Fig. C-terminal tag reveals the expression of Wtf4$^{antidote}$ prior to spore formation.** (**A**) Allele transmission and fertility (assayed via viable spore yield) of 15 diploids with the depicted genotypes. The genotype column shows a cartoon depiction of the relevant genotype. The progeny phenotypes are then shown on the right. For diploids heterozygous at one locus (e.g., Diploid 1), two values are shown (top and bottom) that represent the two possible haploid genotypes. The depictions are not to scale with the location of the loci on the chromosomes. Spores exhibiting both parental phenotypes were considered diploid or aneuploid and were excluded from this table but can be found in S1 Data. The expected values assume Mendelian allele transmission. We used the viable spore yield assay (VSY) to quantify fertility with values normalized to the relevant empty vector control (* = p < 0.05, NS = not significant; G-test for allele transmission, Wilcoxon test for VSY, in comparison to the empty vector control). We compared diploids 3, 4, 5, 6, 8, 10, 12, 13 to control diploid 1 and diploids 7, 9, 11, 14, 15 to control diploid 2. The data for diploids 1, 2, and 12–15 are also depicted in S10A Fig and the data for diploids 1 and 5 are also depicted in Fig 3C. The data from diploids 3, 4, and 14 were previously published in [37]. (**B**) Images of a heterozygous *mCherry-wtf4$^{antidote}$/ade6+* diploid cell and mature ascus. mCherry-Wtf4$^{antidote}$ is shown in magenta in merged images. (**C**) Images of a heterozygous *wtf4$^{antidote}$-mCherry/ura4+* diploid cell and mature ascus. Wtf4$^{antidote}$-mCherry is shown in magenta. (**D**) Images of a heterozygous *wtf4$^{antidote}$-GFP/ade6+* diploid cell and mature ascus. Wtf4$^{antidote}$-GFP is shown in magenta. All Images were taken after 3 days on sporulation media. TL = transmitted light. All scale bars represent 2 μm. Images were taken with the same settings. Not all images are shown at the same brightness and contrast to avoid over saturation of pixels in the brighter images.
(TIF)

**S3 Fig. Wtf4$^{poison}$ is produced prior to spore individualization.** (**A**) Representative images from a Fluorescence Recovery After Photobleaching (FRAP) experiment with mature asci (n = 10) generated from *mCherry-wtf4/ wtf4$^{poison}$-GFP* diploids. (**B**) FRAP of both mCherry

(magenta line) and GFP (cyan line) post bleaching to 0% intensity and recovery quantified over 6 hours. (**C**) Representative images from a FRAP experiment with spores in mature asci (n = 40) generated from *wtf4^antidote^-mCherry/wtf4^poison^*-GFP diploids. (**D**) FRAP of GFP (cyan line) post bleaching to 0% intensity and recovery was quantified over 3 hours. Not all images are shown at the same brightness and contrast to avoid over saturation of pixels in the brighter images.
(TIF)

**S4 Fig. Wtf4^poison^ and Wtf4^antidote^ protein expression time-course in *S. pombe* meiosis.** (**A**) Time course of *wtf4^antidote^-mCherry/wtf4^poison^-GFP* diploid showing the localization of mCherry-Wtf4^antidote^ (magenta in merged images) and Wtf4^poison^-GFP (cyan in merged images) after the indicated times in PM-N media. All scale bars represent 10 μm. These are images of the cell populations sampled for the westerns. (**B**) Western blot of whole cell extracts from cells expressing *wtf4^poison^-GFP* at the times indicated. *Indicates a non-specific band. Two biological replicates are shown on the gel. (**C**) Western blot of whole cell extracts from cells expressing *wtf4^antidote^-mCherry* at the times indicated. (**D**) A replicate gel was stained with Imperial protein stain to show quantity of protein loaded.
(TIF)

**S5 Fig. Wtf4^antidote^-mCherry and mCherry- Wtf4^antidote^ protein expression time-course in *S. pombe* meiosis.** (**A**) Time course of *mCherry- wtf4^antidote^*/EV *and wtf4^antidote^-mCherry*/EV diploids showing the localization of Wtf4^antidote^ (magenta in merged images) after the indicated times in PM-N media. All scale bars represent 10 μm. These are images of the cell populations sampled for the westerns. (**B**) Western blot of whole cell extracts from cells expressing either *mCherry- wtf4^antidote^* or *wtf4^antidote^-mCherry* at the times indicated. The mCh control was isolated from mitotically growing cells expressing constitutive mCherry. (**C**) A replicate gel was stained with Imperial protein stain to show quantity of protein loaded.
(TIF)

**S6 Fig. Wtf4^poison^ and Wtf4^antidote^ protein expression time-course in synchronized *S. pombe* meiosis.** (**A**) Time course showing meiotic progression of h-/h- *pat1.L95G/pat1.L95G* diploid cells that were heterozygous for both *wtf4^poison^-GFP* and *wtf4^antidote^-mCherry* [57] 0–24 hours (top) following the addition of 3-MB-PP1 (25 μM) to induce meiosis. The number of nuclei in DAPI-stained cells was counted at the indicated time points. (**B**) Anti-GFP Western blot of whole cell extracts of cells from the same experiment depicted in **A.** Bands consistent with Wtf4^poison^-GFP and free GFP are highlighted with arrows. *indicates a non-specific band. The GFP and mCherry controls were prepared from sporulated cells expressing the free fluorophores constitutively (SZY2636 and SZY2638; see methods). The DMSO control was prepared from a diploid with the same genotype as the experimental sample, but treated for 24 hours with DMSO instead off 3-MB-PP1. (**C**) Anti-mCherry Western blot on the same membrane as that shown in **B**. Bands consistent with Wtf4^antidote^-mCherry and free mCherry are highlighted with arrows. (**D**) Ponceau staining of the membrane in **B-C**. (**E**) The same protein samples used in the blot in **B-C** were rerun on a new anti-mCherry Western where blotting stringency was increased by the addition of 0.2% Tween 20 to the primary antibody. A band consistent with Wtf4^antidote^-mCherry is observed and is not present in the negative control lanes. (**F**) Ponceau staining of the membrane shown in **E**. The DMSO control sample was erroneously not loaded on this blot.
(TIF)

**S7 Fig. Expression of *wtf4^antidote^* promoter reporters in diploids and asci.** Depictions and images of the two lengths of *wtf4^antidote^* promoters used in this study, *p^antidote^* (**A**) and *p^antidote^*

$^{long}$ (**B**). The images of the two different $p^{antidote}$-*mCherry* reporters / + were from heterozygous diploids and asci. (**C**) Quantification of mCherry fluorescence within heterozygous ($p^{antidote}$-*mCherry* reporter/ +) diploids and asci. At least 25 diploids and 25 asci were quantified per reporter. All images were acquired after 3 days on sporulation media. TL = transmitted light. All scale bars represent 2 μm. Images were taken at the same settings and are shown at the same brightness and contrast for accurate comparison.
(TIF)

**S8 Fig. Mei4 associates with LTRs in meiosis.** Peaks (called by MACS2) from Mei4 ChIP-seq (data from [50]) 4 hours past meiotic induction are compared to LTR regions (identified by BLAST). If the regions overlap, they are shown in the shared region of the Venn diagram. For any ChIP-seq reads that mapped to more than one location, only a single location, chosen at random, was selected.
(TIF)

**S9 Fig. Wtf4$^{poison}$ and Wtf4$^{antidote}$ functions are disrupted when expressed from the p$^{antidote}$ and p$^{poison}$ promoter, respectively.** (**A**) Allele transmission and fertility (assayed via viable spore yield) of 11 diploids of the depicted genotypes. The genotype column shows a cartoon depiction of the relevant genotype. The progeny phenotypes are then shown on the right. For diploids heterozygous at one locus (e.g., diploid 1), two values are shown (top and bottom) that represent the two possible haploid genotypes. For diploids heterozygous at two loci, the loci used are unlinked and should segregate randomly. The depictions are not to scale with the location of the loci on the chromosomes. Spores exhibiting both parental phenotypes were considered diploid or aneuploid and were excluded from this table but can be found in S1 Data. The expected values assume Mendelian allele transmission. (* = p < 0.05, NS = not significant; G-test for allele transmission, Wilcoxon test for VSY, in comparison to the empty vector control). We compared diploids 12, 13, 14, 20 to diploid 1 as the control and diploids 15, 21, 22, 23, 24 to diploid 2 as the control. The data for the control diploids 1–2 and 12–15 are also depicted in S2A Fig. The data for diploid 1 is also in Fig 3C. (**B**) Images of $p^{antidote}$-*wtf4$^{poison}$*-GFP/ade6+, *wtf4$^{antidote}$*-mCherry/ura4+ parental and non-parental ditype asci. $p^{antidote}$-Wtf4$^{poison}$-GFP is shown in cyan and Wtf4$^{antidote}$-mCherry is shown in magenta in merged images. TL = transmitted light. All scale bars represent 2 μm. All images acquired after 2 days on sporulation media. All images are shown at the same brightness and contrast for accurate comparison.
(TIF)

**S1 Video. Time-lapse videos of representative cells shown in Fig 1C and 1E.** Panel A in the video shows a representative *mCherry-wtf4/wtf4$^{poison}$*-GFP diploid undergoing meiosis, captured by the time lapse microscopy we conducted in Fig 1C. mCherry-Wtf4$^{antidote}$ is in magenta and Wtf4$^{poison}$-GFP is in cyan in the composite. Panel B in the video shows a representative *wtf4$^{antidote}$-mcherry/wtf4$^{poison}$*-GFP diploid undergoing meiosis, captured by the time lapse microscopy we conducted in Fig 1E. Wtf4$^{antidote}$-mCherry is in magenta and Wtf4$^{poison}$-GFP is in cyan in the composite. Scale bar represents 4 μm. The videos were captured using the same settings. The videos are not shown at the same brightness and contrast to avoid over saturation of pixels in the brighter frames. Each video is comprised of 115 frames captured at 10 minute intervals.
(MOV)

**S1 Table. Yeast strains used.** Column 1 is the name of strain used, while column 2 refers to the genotype. Columns 3 lists the reference for the yeast strain. If it was made in this study, we also detail how the strain was made in column 4. Column 5 lists the figure(s) in which the

strain was used.
(XLSX)

**S2 Table. Plasmids used.** Column 1 is the name of plasmid used. Column 2 gives a short
description of the plasmid. Columns 3 lists the reference for the plasmid.
(XLSX)

**S3 Table. Oligos Used.** Column 1 is the name of oligo used. Column 2 details the sequence of
the oligo, while column 3 gives a short description.
(XLSX)

**S1 Data. Genetics and Viable Spore Yield (VSY) data for all diploids presented.** Tab 1 is a
summary of all the allele transmission and VSY data for all 22 diploids. Each subsequent tab
contains the raw data for individual crosses. The tab name contains the diploid number, corre-
sponding to the diploid number in the paper, and the two strains crossed to generate the dip-
loid. The top table in each tab lists the VSY data, including the data for at least 3 independent
diploids (A-X) per cross. The number of colonies for the diploid dilutions and the correspond-
ing spore dilutions are listed, as well as the average viable spore yield, standard deviation, and
the p-value when compared to the appropriate control (Wilcoxon test, $^{*}$ = p < 0.05, NS = not
significant). The bottom table contains the allele transmission data. We detail the genotypes of
the two strains crossed to generate the diploid (allele 1 and allele 2) and the transmission fre-
quencies of the two alleles, as well as a control locus. We show the data with and without dis-
omes in the table, but used the data excluding disomes in the figures. We also list the p-values
compared to appropriate control (G-test, $^{*}$ = p < 0.05, NS = not significant).
(XLSX)

**S2 Data. Raw data of image quantification.** Each tab contains the raw data of quantifications
presented in the manuscript. The first tab contains the quantifications depicted in Fig 1D and
1F, where the normalized fluorescence intensity of mCherry and GFP were calculated for dip-
loids undergoing meiosis. Traces for each diploid were plotted as the mean with 95% confi-
dence interval against time. The second tab contains the quantification depicted in Fig 2C and
2F of reporter (promoter-FP) expression in spores. For each ascus, we detail the mean fluores-
cence intensity of the 4 individual spores and the total of the four spores. We then divided
each individual spore by the total to get the percent of total spore fluorescence inside a given
spore. The third tab details the quantification depicted in Fig 4D of percent intensity inside
spores of given fluorescent proteins. We show the percent of fluorescent protein inside spores
in three different genotypes ($p^{poison}$-$wtf4^{antidote}$-GFP/mCh-$wtf4^{poison}$, $p^{poison}$-$wtf4^{antidote}$-GFP/
ade6+, mCh-$wtf4^{poison}$/ade6+). The fourth tab details the Fluorescence Recovery After Photo-
bleaching (FRAP) quantifications depicted in S3B Fig. We normalized the fluorescence inten-
sity before bleaching to 1. Once bleaching was completed, the recovery was recorded every 10
minutes for approximately 6 hours. The fluorescence levels of both mCherry-Wtf4 and
Wtf4$^{poison}$-GFP are recorded for each time point. The fifth tab details the Fluorescence Recov-
ery After Photobleaching (FRAP) quantifications depicted in S3D Fig. The sixth tab details the
quantifications shown in S7C Fig, with the fluorescence intensity inside asci and diploids for
the following constructs: $p^{antidote}$-mCherry, $p^{antidote\ long}$-mCherry, and the empty vector con-
trol. The seventh tab details Fig 5D, the percent intensity inside spores of given fluorescent
proteins. We show the percent of Wtf4$^{poison}$-GFP inside spores in three different genotypes:
$p^{antidote}$-$wtf4^{poison}$-GFP/ade6+, $wtf4^{antidote}$-mCh/ura4+; $wtf4^{poison}$-GFP/ade6+; and $wtf4^{poison}$-
GFP/ade6+, $wtf4^{antidote-}$mCh/ura4+.
(XLSX)

## Acknowledgments

We would like to thank the members of the Zanders lab for their helpful comments on the paper. We thank José Ayté for providing strains. This work was performed to fulfill, in part, requirements for NLN's thesis research in the Graduate School of the Stowers Institute for Medical Research.

## Author Contributions

**Conceptualization:** Nicole L. Nuckolls, Ananya Nidamangala Srinivasa, María Angélica Bravo Núñez, Sarah E. Zanders.

**Data curation:** Nicole L. Nuckolls, Ananya Nidamangala Srinivasa, Anthony C. Mok, Rachel M. Helston, María Angélica Bravo Núñez, Jeffrey J. Lange, Todd J. Gallagher, Chris W. Seidel, Sarah E. Zanders.

**Formal analysis:** Nicole L. Nuckolls, Ananya Nidamangala Srinivasa, Anthony C. Mok, Rachel M. Helston, María Angélica Bravo Núñez, Jeffrey J. Lange, Todd J. Gallagher, Chris W. Seidel, Sarah E. Zanders.

**Funding acquisition:** Nicole L. Nuckolls, María Angélica Bravo Núñez, Sarah E. Zanders.

**Investigation:** Nicole L. Nuckolls, Ananya Nidamangala Srinivasa, Anthony C. Mok, Rachel M. Helston, María Angélica Bravo Núñez, Jeffrey J. Lange, Todd J. Gallagher, Chris W. Seidel.

**Methodology:** Nicole L. Nuckolls, Ananya Nidamangala Srinivasa, Rachel M. Helston, María Angélica Bravo Núñez, Jeffrey J. Lange, Chris W. Seidel, Sarah E. Zanders.

**Project administration:** Sarah E. Zanders.

**Supervision:** Sarah E. Zanders.

**Validation:** Nicole L. Nuckolls, Ananya Nidamangala Srinivasa, Anthony C. Mok, Rachel M. Helston, María Angélica Bravo Núñez, Jeffrey J. Lange, Todd J. Gallagher, Chris W. Seidel.

**Visualization:** Nicole L. Nuckolls, Ananya Nidamangala Srinivasa, Rachel M. Helston, Jeffrey J. Lange, Chris W. Seidel, Sarah E. Zanders.

**Writing – original draft:** Nicole L. Nuckolls.

**Writing – review & editing:** Nicole L. Nuckolls, Ananya Nidamangala Srinivasa, Rachel M. Helston, María Angélica Bravo Núñez, Jeffrey J. Lange, Chris W. Seidel, Sarah E. Zanders.

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
