## [Decision Letter · Decision Letter 0]

8 Nov 2021

Dear Dr Zanders,

Thank you very much for submitting your Research Article entitled 'S. pombe wtf genes use dual transcriptional regulation and selective protein exclusion from spores to cause meiotic drive' to PLOS Genetics.

The manuscript was fully evaluated at the editorial level and by three independent expert peer reviewers. The reviewers appreciated the attention to an important problem, but raised some substantial concerns about the current manuscript. Based on the reviews, we will not be able to accept this version of the manuscript, but we would be willing to review a much-revised version. We cannot, of course, promise publication at that time.

As you can see the reviewers ranged from relatively supportive (Reviewer #2), to expressing  more concern (Reviewer #1), to quite serious concerns (Reviewer #3).  Despite this range of opinions there were common threads throughout the critiques.  The reviewers found the text difficult to read from an organizational perspective.  In addition, they questioned some of the core interpretations of the data, including questions that will require additional experimentation to address.  Reviewer #3 also thought that the novel findings in this study need to be better contextualized in light of your lab's prior work.  The editors agree that better contextualization would be helpful, but we are fine with the reuse of data mentioned by the reviewer since it was properly acknowledged.

If you decide to revise the manuscript for further consideration at PLOS Genetics, please aim to resubmit within the next 60 days, unless it will take extra time to address the concerns of the reviewers, in which case we would appreciate an expected resubmission date by email to plosgenetics@plos.org.

[LINK]

We are sorry that we cannot be more positive about your manuscript at this stage. Please do not hesitate to contact us if you have any concerns or questions.

Yours sincerely,

Gregory P. Copenhaver

Editor-in-Chief

PLOS Genetics

Reviewer's Responses to Questions

**Comments to the Authors:**

Reviewer #1: In fission yeast, wtf genes are meiotic drivers that kill the spores that do not inherit the genes from a heterozygote diploid. Here, the authors focus on wtf4, showing that wtf4-poison and wtf4-antidote are products from a single gene (with wtf4-poison ORF totally included in wtf4-antidote), but with different transcriptional profile. This fact ensures that only spores that have wtf4 gene and express wtf4-antidote (once the spores are formed) will be able to survive.

My major concern with this manuscript is that I do not find any substantial advance over previous manuscripts from the same group (Nuckolls et al, eLIFE, 2017 & 2020); however, there is plenty of room for improvement given the observation that delta-FLEX has no expression of wtf4-poison in meiosis. Could the authors introduce a FLEX element upstream of wtf4-antidote and neutralize wtf4-poison? Can delay wtf4-poison by replacing FLEX element with an element to induce the gene later in meiosis?

Another important point is that all tagged strains should be quantitatively validated on WB on synchronous meiosis (for example, in a pat1-ts strain), determining poison and antidote. There is not a single WB in the whole manuscript

Minor points

Figure 2B and 2D: the figures should extend the sequence, to include contigous sequences.

pg. 10, first paragraph; the authors reference to Nuckolls 2017 to mention that a fully functional wtf4-antidote can be made from a second ATG… However, I could not find the experimental validation for that sentence

pg. 13, first paragraph: the authors wrongly call figures 4D and 4E

pg. 16-17: if the authors want to quantitate the expression of the different promoters, they should do Q-PCR, not follow the expression by Cherry fluorescence.

Reviewer #2: In their article on S. pombe wtf meiotic drivers, Nuckolls et al. investigate the wtf4 spore killer. Previous results have shown that the wtf4 spore killer makes a poison protein and an antidote protein from two distinct but overlapping transcripts. With fluorescent microscopy, Nuckolls et al. provide evidence that the Wtf4 poison is distributed evenly in all four spores of a wtf4/+ heterozygote, while the Wtf4 antidote in enriched in two of the four spores. The authors also show that expression of the Wtf4 antidote can be induced by stress. Finally, they show that expression of the Wtf4 poison is likely controlled by Mei4 (a critical meiotic-stage transcription factor). I found the article to be interesting and the results to be valuable with respect to understanding wtf4-based meiotic drive.

I have some concerns that the authors might want to address in a revised manuscript.

My first concern is with the authors interpretation of the conflicting results with respect to the N- and C-terminal mCherry tags of the Wtf4 antidote. In lines 196-199, the authors write “We conclude that the C-terminal tag reveals the production of a Wtf4 protein population that is not apparent with the N-terminally tagged allele. Although the nature of the additional protein is not clear, it could be produced using the second translational start site found in exon 1 (codon 12)”. I find this explanation to be confusing. Are the authors proposing that the beginning of antidote exon 1 serves as an internal ribosome entry site? Can they clarify their explanation and cite a similar example where a fluorescent protein coding sequence was bypassed in a transgene? If not, perhaps it would be useful to examine the result of fusing mCherry to the second translational start site.

My second concern is that the authors do not consider the roles of 5’ UTRs while interpreting their results. For example, the antidote promoter (i.e., the 285 bp of sequence found immediately upstream of the antidote start codon) should produce an mRNA with an antidote 5’UTR while the poison promoter (i.e., the 230 bp of sequence found immediately upstream of the poison start codon) should produce an mRNA with a poison 5’ UTR. These UTRs should also be present in the mCherry and GFP mRNAs expressed by the Pantidote-mCherry and Ppoison_GFP constructs. Could the antidote and poison 5’ UTRs be major controllers of when poison and antidote are expressed during ascus development and where the proteins are found within asci? If so, can the authors shed light on how this could influence the interpretation of their results?

My third concern deals with transport of proteins between delimited spores of an ascus. My interpretation of the model as presented in the manuscript is that all poison inside non-wtf4 spores must get into these spores before they are delimited. But this is based on the assumption that the poison cannot be transferred from wtf4 spores to non-wtf4 spores after spore delimitation. Are the authors sure that the poison cannot be exported from wtf4 spores after spore delimitation?

My final concern is with the Mei4 findings. While I agree with the authors that Mei4 likely controls poison expression, the claim in lines 35 to 36 is too strong. For example, the authors state that “we show that the Mei4 transcription factor, a master regulator of meiosis, controls the expression of the wtf4 poison transcript”. Currently, this claim is mostly based on the inability of wtf4 to drive when a putative Mei4 binding site in the poison promoter is deleted. I think this a claim of this magnitude should be supported with biochemical experiments. Alternately, the authors could simply improve the manuscript by changing the text to something like “our results strongly suggest that Mei4 controls…”.

Other comments:

Line 120 – please check formatting of loannoni citation

Line 126 – please confirm text, it currently suggests that S. kambucha is a subspecies of S. pombe, is this true?

Line 131, Figure 1A, etc. I think that it is standard practice to consider anything that remains in a transcript after intron removal to be part of an exon.

Line 473 – please check that the statement is accurate. For example, why “must” all spores be exposed to a lethal dose, couldn’t only the killed spores be exposed to a lethal dose?

Line 1114 – please check bold font in ref 94.

Reviewer #3: This group has enjoyed a series of high quality and informative papers on this very interesting topic over the last several years. This manuscript unfortunately does not represent a continuation of that trend. There is a lot of redundancy with previous papers, to the point of showing replicate data from a previously published paper (Fig 1C, which to the authors credit, they point out) More importantly, this manuscript suffers from some lower quality data which poises some problems for the interpretations. The bigger problem is conceptual. There is no good reason to presume that wtf-poison needs to be opposed during early meiosis as the authors put forth as a guiding premise. The period of time in which poison is unopposed during meiosis is transient, and the fact that wtf-poison expression is toxic when chronically expressed in mitotic cells does not provide a sound rationale for the presumption of wtf-antidote function during early meiosis. The authors are able to engineer an antidote protein allele leading to its accumulation in the cytosol of meiotic cells which bolstered their premise. The problem is that the data uphold the conclusion that the intronic sequences separating exons 1 and 2 contains information that represses transcription upstream of exon 1. This could be simply explained by a transcriptional interference mechanisms whereby Mei4 blocks elongation of the longer transcript, though there are numerous other possibilities. Accordingly, the image quality of the pictures of meiotic cells expressing the wtf-antidote-mCherry allele are poor. It is pointed out in the Fig 1 legend that imaging conditions are not uniform throughout, making it very challenging for the reader to make an apples to apples comparison in order to assess the conclusions. One additional way through this would be to provide some other means of quantification of relative abundances, such as western blotting. Whatever the case may be, the relative amounts of antidote accumulating in the cytosol during early meiosis and in the spores following the completion of meiosis are dramatically different when comparing the mCherry-wtf4 vs wtf4-mCherry-antidote. Independent of the "apples to apples" comparison problem of mCherry-wtf4 vs wtf4-antidote-mCherry, this internal comparison of relative accumulation within each experiment informative and poses a challenge to much of the manuscript: relative to antidote accumulation in the developed spores, wtf-antidote-mCherry produces dramatic accumulation of wtf4-antidote in early meiotic cells in marked contrast to that observed using the more native mCherry-wtf4 allele. This is not adequately explained, and as described above, it seems like the best explanation for this is the that the intronic sequences removed from the wtf4-antidote-mcherry allele lead to de-repression of transcription of the longer antidote-encoding transcript. Thus, the premise for the rest of the paper is dubious.

Additional comments.

This work would really benefit from the use of Northern blots to display the two transcripts and their kinetics during meiosis. I realize the existence of the 2 transcripts is now well supported from analysis of genomic data sets and additional experiments. However, given the centrality of the role of the two wtf4 transcript isoforms, and the use now of so many constructs deleting and/or adding information to these transcripts, it is my view that Northern blotting is an essential complement to the authors approaches. For example, does deletion of intron 1 (or just the Mei4 target sequence) lead to de-repressed transcription from the longer isoform?

For 2G. Is the poison-promoter-GFP seen in spores really representative of “expression” within them, or just inheritance of GFP from reporter expression during meiosis?

3A already shown in 1D. exact same result.

3B. wtf-antidote under a wtf-poison promoter. exon 1 sequences (RNA or its encoded protein), prevent poison from being imported into spores. Not explained well by authors.

Authors argue that promoter-reporter constructs recapiluate 1D, but this is not really the case. The wtf4-antidote-mcherry reporter shows much more convincing expression in meiotic cells than the antidote-promoter-GFP reporter.

Why does diploid 7 show a reduced frequency of G418/Hyg sensitive spores?

**Have all data underlying the figures and results presented in the manuscript been provided?**

Reviewer #1: None

Reviewer #2: Yes

Reviewer #3: Yes

PLOS authors have the option to publish the peer review history of their article (what does this mean?). If published, this will include your full peer review and any attached files.

Reviewer #1: No

Reviewer #2: No

Reviewer #3: No

---

## [Decision Letter · Decision Letter 1]

31 May 2022

Dear Dr Zanders,

Thank you very much for submitting your Research Article entitled 'S. pombe wtf genes use dual transcriptional regulation and selective protein exclusion from spores to cause meiotic drive' to PLOS Genetics.

The manuscript was fully evaluated at the editorial level and by independent peer reviewers. Reviewers #2 and #3 are now satisfied, but reviewer #1 continues to have concerns centered around quantifying the poison and antidote.  Reviewer #1 has the following 5 comments which I provide guidance for.  Once you address these I will be able to render a final editorial decision without further external review.

1. Attempting western blot quantification with highly synchronous background (like pat1). I ask that you consider trying to address this concern. Like the reviewer, I understand that using a different background from the other experiments is sub-optimal, but readers could be alerted to that caveat in the text.

2. Please amend the Materials and Methods to include the requested information.

3. Please mend the indicated figures to include MW information in figures S5D and S6C.

4. I downloaded the original files for figures S5 and S6 and I think the quality of the figures is acceptable (and the other reviewers did not express any concern) - responding to this concern is not necessary.

5. I see what the reviewer is referring to, but I don't know enough about pombe morphology to know if it is a problem. At a minimum, please comment on this in your response letter. If you think there is an underlying issue that would alter interpretation of the experiments a more robust response may be warranted.

[LINK]

Yours sincerely,

Gregory P Copenhaver

Editor-in-Chief

PLOS Genetics

Reviewer's Responses to Questions

**Comments to the Authors:**

Reviewer #1: This is a revised version of a manuscript previously submitted to PLoS Genetics. I can appreciate the efforts that the authors have done to improve the manuscript, following the comments from the reviewers and the suggestions from the editor, including adding new figures and entirely re-writing the manuscript.

Regarding my previous concerns, I am satisfied with the replies from the authors and with the specific changes that they have introduced in the manuscript, with one exception that I believe that should be corrected to improve the manuscript. I have serious concerns with the WB that they have included in the supplementary information (Figures S5 and S6):

- I understand that all the meiosis in the manuscript are in a “wild type” background, and that could be a reason for not using an “artificial” pat1-driven meiosis. But in view of the WB that the authors are showing, I think that they should use a pat1-ts background for the only WB experiments; pat1-driven meiosis allows a highly synchronous meiosis (much better than a h+/h- diploid meiosis) and might allow to detect some of the tagged proteins that so far they are not detecting (i.e. antidote-mCherry, Fig S5; mCherry-Antidote and Antidote-mCherry, Fig S6).

- What kind of protein extracts are the authors preparing? It is not indicated in the Materials and Methods section, neither the buffer in which cells are broken. If the authors suspect a problem with stability of the proteins (as it happens with many proteins expressed during meiosis), they should prepare TCA extracts; if there is a problem of solubility, then boiled extracts (running membrane fraction in the electrophoresis)

- Molecular weight markers are not indicated in Fig S5D and S6C; in Figures S5BC and S6B, only 4 markers are indicated

- The quality of fluorescence in these two figures is poor, compared to other figures in the same manuscript.

- Fig S6: why cells are so different in size at T=0 hours in the two top panels, compared with the bottom panel or with other figures? Cells with the empty vector or with the mCh-antidote are elongated in this specific experiment, indicating that probably there is something wrong with these two cultures

Reviewer #2: In their revised article on S. pombe wtf meiotic drivers, Nuckolls et al. investigate the wtf4 spore killer. The authors have satisfactorily addressed all of my questions concerning their initial version of the manuscript. I believe the current version of the manuscript represents a significant and valuable contribution to our understanding of spore killer-based gene drivers.

Minor suggestions:

Line 120 - maybe "diver" should be "driver"

Line 187 - maybe delete the first occurrence of "that"

Line 199 - maybe change to "…lacks introns, and thus…"

Line 230 - maybe delete "from at timepoints" and "in timepoints"

Reviewer #3: The manuscript is greatly improved and many of my comments have been addressed.

**Have all data underlying the figures and results presented in the manuscript been provided?**

Reviewer #1: None

Reviewer #2: Yes

Reviewer #3: Yes

PLOS authors have the option to publish the peer review history of their article (what does this mean?). If published, this will include your full peer review and any attached files.

Reviewer #1: No

Reviewer #2: No

Reviewer #3: No

---

## [Editor Report · Decision Letter 2]

15 Nov 2022

Dear Dr Zanders,

We are pleased to inform you that your manuscript entitled "S. pombe wtf drivers use dual transcriptional regulation and selective protein exclusion from spores to cause meiotic drive" has been editorially accepted for publication in PLOS Genetics. Congratulations!

Yours sincerely,

Gregory P. Copenhaver

Editor-in-Chief

PLOS Genetics

Comments from the reviewers (if applicable):

**Data Deposition**

http://datadryad.org/submit?journalID=pgenetics&manu=PGENETICS-D-21-01298R2

**Press Queries**

---

## [Editor Report · Acceptance letter]

2 Dec 2022

PGENETICS-D-21-01298R2 

S. pombe wtf drivers use dual transcriptional regulation and selective protein exclusion from spores to cause meiotic drive 

Dear Dr Zanders, 

We are pleased to inform you that your manuscript entitled "S. pombe wtf drivers use dual transcriptional regulation and selective protein exclusion from spores to cause meiotic drive" has been formally accepted for publication in PLOS Genetics! Your manuscript is now with our production department and you will be notified of the publication date in due course.

With kind regards,

Zsofi Zombor

PLOS Genetics

On behalf of:
